# TrkB-expressing paraventricular hypothalamic neurons suppress appetite through multiple neurocircuits

Juan Ji An[1], Clint E. Kinney[1], Ji-Wei Tan[1], Guey-Ying Liao[1], Eric J. Kremer[2] & Baoji Xu [1✉]

The TrkB receptor is critical for the control of energy balance, as mutations in its gene (*NTRK2*) lead to hyperphagia and severe obesity. The main neural substrate mediating the appetite-suppressing activity of TrkB, however, remains unknown. Here, we demonstrate that selective *Ntrk2* deletion within paraventricular hypothalamus (PVH) leads to severe hyperphagic obesity. Furthermore, chemogenetic activation or inhibition of TrkB-expressing PVH (PVH^TrkB) neurons suppresses or increases food intake, respectively. PVH^TrkB neurons project to multiple brain regions, including ventromedial hypothalamus (VMH) and lateral parabrachial nucleus (LPBN). We find that PVH^TrkB neurons projecting to LPBN are distinct from those to VMH, yet *Ntrk2* deletion in PVH neurons projecting to either VMH or LPBN results in hyperphagia and obesity. Additionally, TrkB activation with BDNF increases firing of these PVH neurons. Therefore, TrkB signaling is a key regulator of a previously uncharacterized neuronal population within the PVH that impinges upon multiple circuits to govern appetite.

[1] Department of Neuroscience, The Scripps Research Institute Florida, Jupiter, FL 33458, USA. [2] Institut de Génétique Moléculaire de Montpellier, University of Montpellier, CNRS, Montpellier, France. ✉email: bxu@scripps.edu

The regulation of food intake is essential for survival in all organisms, and its dysregulation not only affects survival but also causes health problems such as obesity. Several ligand-receptor pairs play a crucial role in the central regulation of energy balance. They include leptin and the leptin receptor, alpha-melanocyte-stimulating hormone (αMSH) and the melanocortin-4 receptor (MC4R), and brain-derived neurotrophic factor (BDNF) and the TrkB receptor. Defects in any of these three ligand-receptor pairs lead to severe obesity in humans[1–5] and mice[6–11]. Furthermore, several common single nucleotide polymorphisms in the *MC4R* and *BDNF* genes have been found to be associated with increased body mass index in large-scale genome-wide association studies[12–15]. Remarkable progress has been made in elucidation of neural circuits that mediate the effects of leptin and αMSH on energy balance[16]. By comparison, much less is known about the neural substrate through which the BDNF-TrkB pathway regulates appetite and energy expenditure.

BDNF is well known for its role in neuronal development and synaptic function[17–19]. It signals through the binding of two distinct classes of receptor proteins: the tropomyosin receptor kinase B (TrkB) and the p75 neurotrophin receptor (p75[NTR]). Mutations in the *NTRK2* gene, which codes for TrkB, lead to severe obesity in mice and humans[4,11]. Conversely, no obesity phenotype has been observed in mice lacking the p75[NTR] receptor[20]. In fact, mice lacking the p75[NTR] receptor are protected from obesity induced by high-fat diet and remain lean due to increased energy expenditure[21]. Therefore, BDNF should signal through TrkB to regulate energy balance. The paraventricular hypothalamus (PVH) and the ventromedial hypothalamus (VMH) have been identified as two important brain areas that express BDNF to suppress food intake and promote energy expenditure[22–24]. However, it remains unclear in which population of hypothalamic neurons TrkB is required for the control of energy balance and how these TrkB-expressing neurons regulate food intake and/or energy expenditure. Our recent work found that *Ntrk2* deletion in the dorsomedial hypothalamus (DMH) led to modest hyperphagia and obesity[25], but to a much lesser extent in comparison with *Ntrk2* hypomorphic mice in which *Ntrk2* expression is reduced to a quarter of the normal amount throughout the body[11], indicating that the main neural substrate underlying the action of TrkB signaling on energy balance remains to be identified.

Here we report that TrkB-expressing PVH (PVH[TrkB]) neurons are distinct from neurons that express BDNF, MC4R, glucagon-like peptide 1 receptor (GLP1R), or prodynorphin (PDYN) in the PVH, the four subtypes of neurons that are known to suppress appetite[22,26–28]. We find that chemogenetic inhibition of PVH[TrkB] neurons greatly increases food intake, whereas chemogenetic activation of these neurons drastically reduces food intake. Furthermore, we find that deletion of the *Ntrk2* gene in the PVH leads to severe hyperphagic obesity. We then use a projection-specific gene deletion approach to reveal that PVH[TrkB] neurons project to the VMH and the lateral parabrachial nucleus (LPBN) to suppress food intake. These studies not only identify the main neural substrate by which TrkB signaling regulates energy balance, but also implicates the PVH → VMH neurocircuit in the control of appetite.

## Results

**Many TrkB neurons in the PVH are distinct from known ones.** The PVH is a heterogeneous brain structure with many different cell types[29,30]. At least 11 types of PVH neurons can be molecularly defined by the expression of BDNF, corticotropin-releasing hormone (CRH), growth hormone-releasing hormone (GHRH), MC4R, oxytocin, PDYN, GLP1R, somatostatin, tyrosine hydroxylase, thyrotropin-releasing hormone (TRH), and vasopressin[22,26,27,31]. We previously found that approximately 5% PVH[TrkB] neurons expressed BDNF[22]. We sought to determine whether PVH[TrkB] neurons belong to other defined neuronal populations.

TrkB antibodies do not mark cell bodies well in brain sections, because TrkB is distributed to the membrane of both cell bodies and processes. We employed an *Ntrk2*[CreER/+] mouse strain[32], in which the Cre-ERT2 sequence is inserted into the *Ntrk2* locus immediately after the first coding ATG, for colocalization studies. We crossed *Ntrk2*[CreER/+] mice to *Rosa26*[Ai9/+] mice[33] to generate *Ntrk2*[CreER/+];*Rosa26*[Ai9/+] mice, in which TrkB-expressing neurons should express tdTomato after tamoxifen treatment. Since the kinase-lacking truncated TrkB is expressed in astrocytes[34], tdTomato should also label astrocytes. We found that 83% (1206/1449) of PVH cells containing *Ntrk2* mRNA were positive for tdTomato mRNA, whereas 86% (1206/1404) of PVH cells containing tdTomato mRNA were positive for *Ntrk2* mRNA (Fig. 1a). Thus, tdTomato efficiently and faithfully marks TrkB-expressing cells in tamoxifen-treated *Ntrk2*[CreER/+];*Rosa26*[Ai9/+] mice. We detected many tdTomato-labelled TrkB cells in the PVH (Fig. 1b), which are either neurons that are positive for the neuronal marker NeuN (Supplementary Fig. 1a) or astrocytes that are positive for glial fibrillary acidic protein (Supplementary Fig. 1b). We noticed that there was a high colocalization between TrkB and oxytocin in the PVH (18% TrkB neurons express oxytocin and 51% oxytocin neurons express TrkB; Fig. 1c) and between TrkB and GLP1R (14% cells positive for *Ntrk2* mRNA express *Glp1r* mRNA and 37% cells positive for *Glp1r* mRNA express *Ntrk2* mRNA; Fig. 1d). In addition, a small number of PVH[TrkB] neurons express TRH (2% TrkB neurons express TRH and 12% TRH neurons express TrkB; Supplementary Fig. 1c), PDYN (6% TrkB neurons express PDYN and 17% PDYN neurons express TrkB; Fig. 1e), somatostatin (2% TrkB neurons express somatostatin and 16% somatostatin neurons express TrkB; Supplementary Fig. 1d) and GHRH (5% TrkB neurons express GHRH and 30% GHRH neurons express TrkB; Supplementary Fig. 1e). Note that higher percentages of PVH[TrkB] neurons are positive for these markers in the shown images (Fig. 1 and Supplementary Fig. 1), because the images were taken from the area where these markers are expressed. Very few PVH[TrkB] neurons express CRH (Supplementary Fig. 2a), tyrosine hydroxylase (Supplementary Fig. 2b), or vasopressin (Supplementary Fig. 2c). We also examined colocalization of TrkB with MC4R in the PVH using the BAC *Mc4r-EGFP* mouse strain[35]. We only detected low colocalization of TrkB with MC4R in the PVH of tamoxifen-treated *Ntrk2*[CreER/+];*Rosa26*[Ai9/+];*Mc4r-EGFP* mice (3% TrkB neurons express MC4R and 10% MC4R neurons express TrkB; Fig. 1f). Because there are overlaps in expression between the aforementioned markers in the PVH[22,28,36], the sum of TrkB neurons expressing these markers should be significantly lower than 51% of total PVH[TrkB] neurons (Supplementary Fig. 2d). These results indicate that the majority of PVH[TrkB] neurons are distinct from previously defined neuronal populations.

**Deletion of the *Ntrk2* gene with Sim1-Cre leads to obesity.** We previously reported that *Ntrk2* deletion in several hypothalamic areas using the Rgs9-Cre led to obesity[37]; however, the obesity phenotype was not as severe as what we observed in *Ntrk2* hypomorphic mice that express TrkB at ~25% of the normal amount throughout the body[11,37]. Given that Rgs9-Cre only deletes *Ntrk2* in a small number of PVH neurons[37] and that stereotaxic injection of recombinant BDNF into the PVH reduces

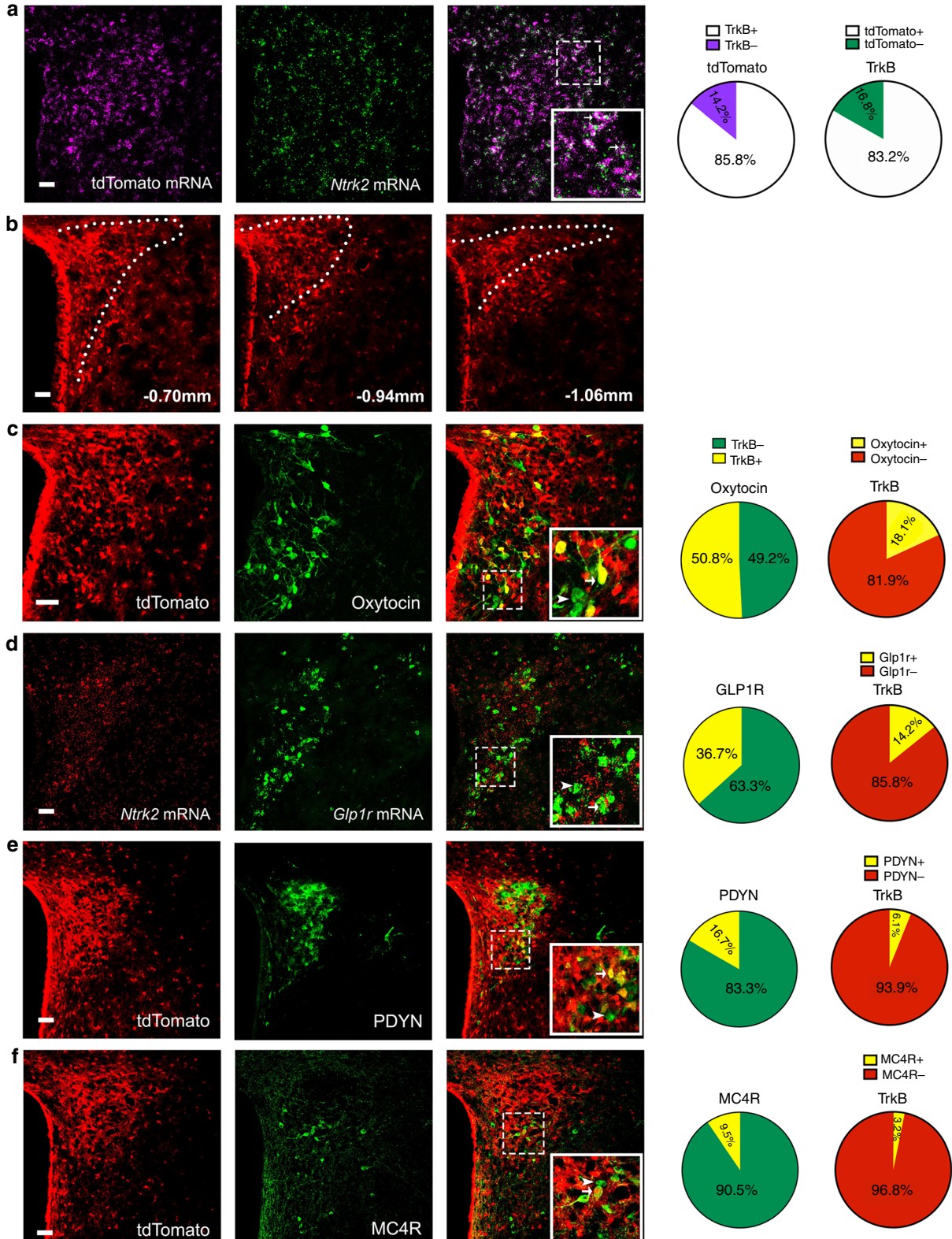

**Fig. 1 Co-expression of TrkB with other neuronal markers in the PVH.** TrkB-expressing cells are marked by tdTomato in tamoxifen-treated *Ntrk2*$^{CreER/+}$; *Rosa26*$^{Ai9/+}$ mice. **a** Co-expression of tdTomato mRNA with *Ntrk2* mRNA. **b** Distribution of TrkB-expressing cells in the PVH at three rostral-caudal positions. **c–e** Co-expression of TrkB with oxytocin, GLP1R, and PDYN. **f** Co-expression of tdTomato and EGFP in *Ntrk2*$^{CreER/+}$; *Rosa26*$^{Ai9/+}$;*Mc4r-EGFP* mice. Arrows denote representative neurons expressing both TrkB and a PVH marker. Scale bars are 50 µm long. Source data are provided as a Source Data file.

food intake and increases energy expenditure[38,39], this observation raises the possibility that TrkB expressed in the PVH (PVH TrkB hereafter) plays an important role in the control of energy balance. We tested this possibility by deleting *Ntrk2* in the PVH.

We tested whether Sim1-Cre[40] is suitable for abolishment of *Ntrk2* gene expression in the PVH using Sim1-Cre;*Ntrk2*^fBZ/+^ mice. *Ntrk2*^fBZ^ is a floxed allele and contains an inserted sequence encoding tau-β-galactosidase[41,42]. It expresses tau-β-galactosidase under the control of the endogenous *Ntrk2* promoter after its floxed sequence is deleted by Cre recombinase. Therefore, tau-β-galactosidase expression in Sim1-Cre;*Ntrk2*^fBZ/+^ mice indicates where the Sim1-Cre transgene could be used to abolish TrkB expression. We detected many cells expressing β-galactosidase in the PVH (Fig. 2a), suggesting that Sim1-Cre could be used to effectively abolish TrkB expression in the PVH. In addition to the PVH, Sim1-Cre could also abolish TrkB expression in some neurons of other brain regions, including the cerebral cortex, hippocampus, amygdala, thalamus, and DMH (Supplementary Fig. 3a–f).

We crossed Sim1-Cre mice to *Ntrk2*^lox/lox^ mice, which express a normal amount of TrkB in the absence of Cre[43], to produce *Ntrk2*^lox/lox^ (control) and Sim1-Cre;*Ntrk2*^lox/lox^ (mutant) mice. In situ hybridization confirmed that *Ntrk2* expression was abolished in the PVH of Sim1-Cre;*Ntrk2*^lox/lox^ mice (Fig. 2b). Male and female Sim1-Cre;*Ntrk2*^lox/lox^ mice at 16 weeks of age were 45 and 49% heavier than sex-matched control mice, respectively (Fig. 2c, d). The Sim1-Cre;*Ntrk2*^lox/lox^ mice also had a longer body (Fig. 2e) and larger adipose tissues relative to control mice (Supplementary Fig. 3g, h). These results indicate that Sim1-Cre; *Ntrk2*^lox/lox^ mice develop obesity.

At 8 weeks of age, female Sim1-Cre;*Ntrk2*^lox/lox^ mice daily consumed 24% more food than control mice (Fig. 2f). We employed indirect calorimetry to estimate energy expenditure in mice at 6 weeks of age when control and Sim1-Cre;*Ntrk2*^lox/lox^ mice had comparable body weights (Fig. 2c). Control and Sim1-Cre;*Ntrk2*^lox/lox^ mice had comparable oxygen consumption ($VO_2$) during the light period when mice are mostly at sleep (Fig. 2g, h), suggesting Sim1-Cre;*Ntrk2*^lox/lox^ mice have a relatively normal basal metabolic rate. During the dark period, $VO_2$ and locomotor activity in Sim1-Cre;*Ntrk2*^lox/lox^ mice were reduced by 15 and 51%, respectively, compared with control mice (Fig. 2g–i). It is unlikely that the reduction in $VO_2$ is secondary to the small, but statistically insignificant, excess weight gain in mutant mice, because the reduction did not occur during the light period (Fig. 2h). These results indicate that TrkB signaling in Sim1-Cre cells promote physical activity and energy expenditure. Collectively, our results indicate that Sim1-Cre;*Ntrk2*^lox/lox^ mice develop obesity due to increased food intake and reduced energy expenditure.

***Ntrk2* deletion in the adult PVH led to hyperphagic obesity.** Sim1-Cre is expressed in other brain regions in addition to the PVH[40] (Supplementary Fig. 3a–f), so the obesity phenotype in Sim1-Cre;*Ntrk2*^lox/lox^ mice might result from *Ntrk2* deletion in non-PVH neurons or cells. To validate a role for PVH TrkB in the regulation of energy balance, we stereotaxically injected neuron-tropic AAV2, either AAV2-CMV-Cre-GFP or AAV2-CMV-GFP, into the PVH of 8-week-old female *Ntrk2*^lox/lox^ mice (Fig. 3a). Injection of AAV2-CMV-Cre-GFP selectively abolished *Ntrk2* expression in the PVH (Fig. 3a2, a3). Mice injected with AAV2-CMV-Cre-GFP are divided into two groups based on the extent of AAV infection in the PVH: AAV2-CMV-Cre-GFP (missed) and AAV2-CMV-Cre-GFP (hit) (Supplementary Table 1). Mice in the AAV2-CMV-Cre-GFP (hit) group developed severe obesity quickly, compared to mice in the AAV2-CMV-GFP group and the

AAV2-CMV-Cre-GFP (Missed) group (Fig. 3b, c). The severity of obesity in AAV-Cre-injected mice had a good correlation with the extent of AAV infection in the PVH (Fig. 3d), but not along needle tracks or in other hypothalamic areas (Supplementary Table 1). These results show that PVH TrkB is critical for the control of energy balance.

Mice in the AAV2-CMV-Cre-GFP (hit) group displayed marked hyperphagia just two weeks after AAV injection and remained hyperphagic 10 weeks after injection (Fig. 3e). These mice had energy expenditure and respiratory exchange ratio comparable to control mice (Fig. 3f, h), whereas their ambulatory activity was trending lower 2 weeks after injection and was reduced 10 weeks after injection (Fig. 3g). These results indicate that PVH TrkB regulates energy balance mainly by suppressing appetite.

We noticed that the extent of AAV infection along a needle track is correlated with injection volume. We sought to further verify the role of PVH TrkB in the control of appetite by injecting female *Ntrk2*^lox/lox^ mice with a smaller volume of AAV into the PVH to minimize infection in needle tracks and other hypothalamic nuclei. We detected very few AAV-transduced neurons in brain areas other than the PVH (mostly detected in Bregma −0.7 to −0.94 mm) in this batch of mice (Supplementary Fig. 4a1–a3). In the PVH, *Ntrk2* gene deletion did not cause neuronal loss during the course of the experiment (Supplementary Fig. 4a4–a6). Mice injected with AAV2-CMV-Cre-GFP displayed the same phenotypes as we observed in the first batch of mice, i.e. severe hyperphagic obesity, increased linear growth and reduced ambulatory activity (Supplementary Fig. 4b–g). Interestingly, even unilateral *Ntrk2* gene deletion was sufficient to cause significant obesity (Supplementary Fig. 4b), indicating that normal energy homeostasis needs full-strength TrkB signaling in PVH neurons that control appetite. Furthermore, *Ntrk2* gene deletion in the PVH also led to obesity in male mice (Supplementary Fig. 4h). We did not detect any toxic effect on PVH neurons from AAV transduction, as bilateral injection of AAV2-CMV-Cre-GFP into the PVH of wild-type mice did not alter body weight (Supplementary Fig. 4i).

Although studies suggest that MC4R signaling interacts with BDNF-TrkB signaling in the context of body weight regulation[11,44], it is unlikely that *Ntrk2* deletion in the PVH leads to obesity through the MC4R because the deletion did not reduce levels of *Mc4r* mRNA (Supplementary Fig. 4j, k). We also investigated whether PVH TrkB controls energy balance by regulating the function of oxytocin neurons, as many oxytocin neurons express TrkB (Fig. 1c) and oxytocin neurons may play a central role in the control of appetite[45–49]. We deleted the *Ntrk2* gene in oxytocin neurons using the *Oxt*^Cre^ mouse strain[50] (Supplementary Fig. 5a) and found that the deletion did not affect body weight in mice of either sex (Supplementary Fig. 5b, c).

Collectively, these selective gene deletion experiments show that signaling through the TrkB receptor potently suppresses appetite through non-oxytocin-expressing PVH neurons. They also indicate that TrkB signaling in the PVH somewhat promotes locomotion without significantly altering energy expenditure.

**PVH^TrkB^ neurons bidirectionally regulate food intake.** As BDNF potently potentiates neuronal growth and synaptic function[19], *Ntrk2* deletion would impair the function of PVH^TrkB^ neurons. The observation that *Ntrk2* deletion in the PVH leads to hyperphagic obesity would implicate PVH^TrkB^ neurons in the acute regulation of food intake. We tested this hypothesis by manipulating the activity of PVH^TrkB^ neurons using designer receptors exclusively activated by designer drugs (DREADDs)

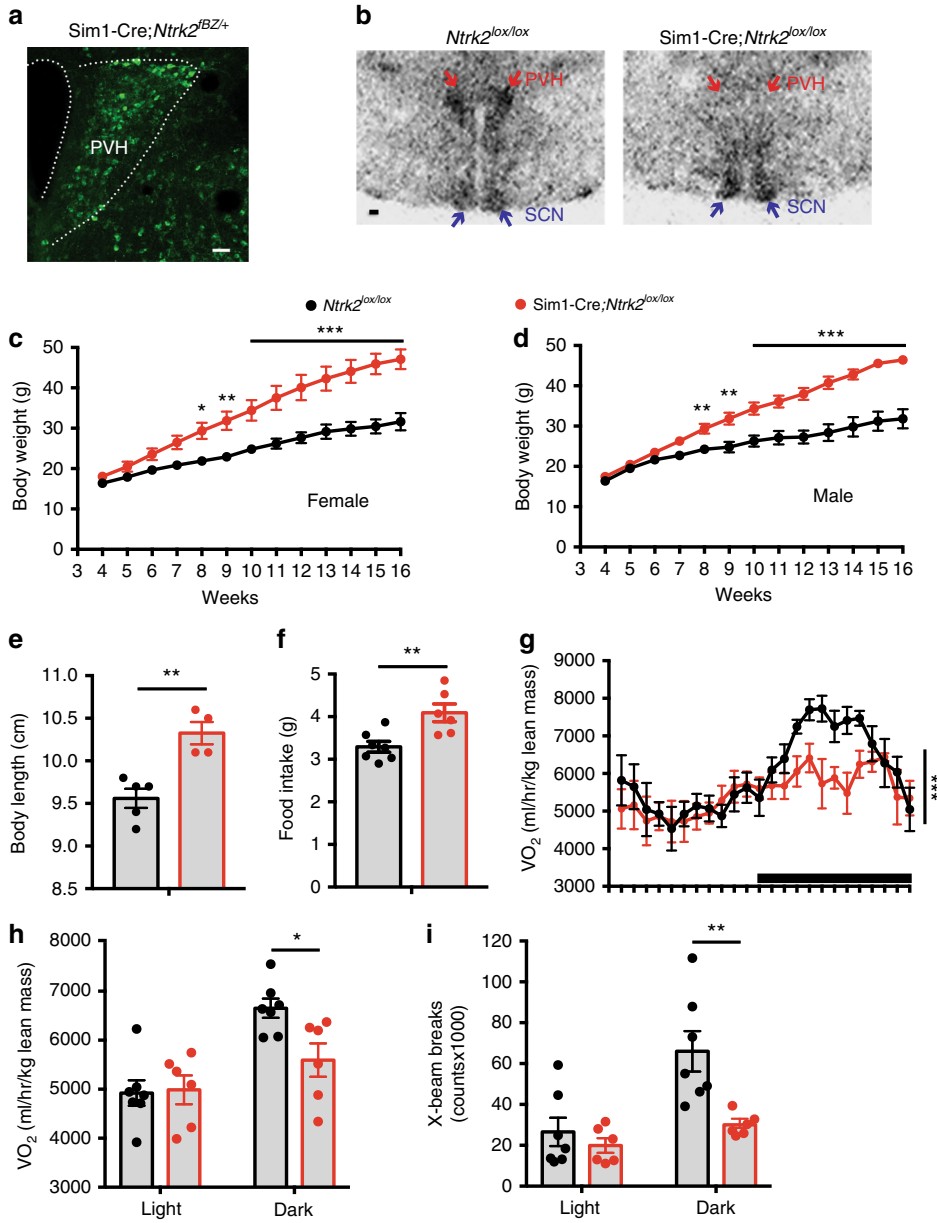

**Fig. 2 Deletion of the _Ntrk2_ gene in Sim1-Cre cells led to obesity. a** β-galactosidase immunohistochemistry of Sim1-Cre;_Ntrk2$^{fBZ/+}$_ mice. Scale bar, 50 μm. **b** In situ hybridization of brain sections from 4-month-old _Ntrk2$^{lox/lox}$_ mice (control) and Sim1-Cre;_Ntrk2$^{lox/lox}$_ mice (mutant) reveals that _Ntrk2_ gene expression was abolished in the PVH of mutant mice. Arrows denote the PVH and the suprachiasmatic nucleus (SCN). Scale bar, 200 μm. **c** Body weight of female control and mutant mice. $n = 5$ mice per genotype. Two-way ANOVA with post hoc Bonferroni multiple comparisons; $F_{(1, 104)} = 163.7$, $P < 0.0001$ for genotype; *$P < 0.05$, **$P < 0.01$, and ***$P < 0.001$. **d** Body weight of male control and mutant mice. $n = 4$ controls and six mutants. Two-way ANOVA with post hoc Bonferroni multiple comparisons; $F_{(1, 104)} = 193.5$, $P < 0.0001$ for genotype; **$P < 0.01$ and ***$P < 0.001$. **e** Body length of 8-week-old female control and mutant mice. $n = 5$ controls and four mutants. Two-tailed unpaired $t$ test; **$P = 0.003$. **f** Daily food intake of 8-week-old female control and mutant mice. $n = 7$ controls and six mutants. Two-tailed unpaired $t$ test; **$P = 0.006$. **g** Distribution of oxygen consumption (VO$_2$) over a 24-h period in 6-week-old female mice. The black bar above $x$ axis indicates the dark cycle. $n = 7$ controls and six mutants. Two-way ANOVA for genotype; $F_{(1, 264)} = 14.54$, ***$P = 0.0002$. **h** VO$_2$ of 6-week-old female mice during the light and dark cycles. $n = 7$ controls and six mutants. Two-tailed unpaired $t$ test; *$P = 0.017$. **i** Locomotor activity of 6-week-old female mice during the light and dark cycles. $n = 7$ controls and six mutants. Two-tailed unpaired $t$ test; **$P = 0.007$. Error bars indicate SEM. Source data are provided as a Source Data file.

and their ligand clozapine N-oxide (CNO)[51,52]. We expressed stimulatory DREADD hM3D(Gq), inhibitory DREADD hM4D (Gi), or control mCherry in PVH$^{TrkB}$ neurons by injecting Cre-dependent AAV2-hSyn-DIO-hM3D(Gq)-mCherry, AAV2-hSyn-DIO-hM4D(Gi)-mCherry, or AAV2-hSyn-DIO-mCherry into the PVH of _Ntrk2$^{CreER/+}$_ mice. We subsequently induced the expression of the injected AAV vectors in PVH$^{TrkB}$ neurons by treating the mice with tamoxifen (Fig. 4a, b).

Administration of CNO did not alter food intake of _Ntrk2$^{CreER/+}$_ mice expressing mCherry during either the light or dark cycle or after fasting (Fig. 4c–f), indicating that CNO does not have any detectable nonspecific effect on feeding. Administration of CNO greatly increased food intake of _Ntrk2$^{CreER/+}$_ mice expressing hM4D(Gi)-mCherry during the light cycle when mice do not normally eat much food, compared with vehicle administration (Fig. 4c). This result indicates that PVH$^{TrkB}$

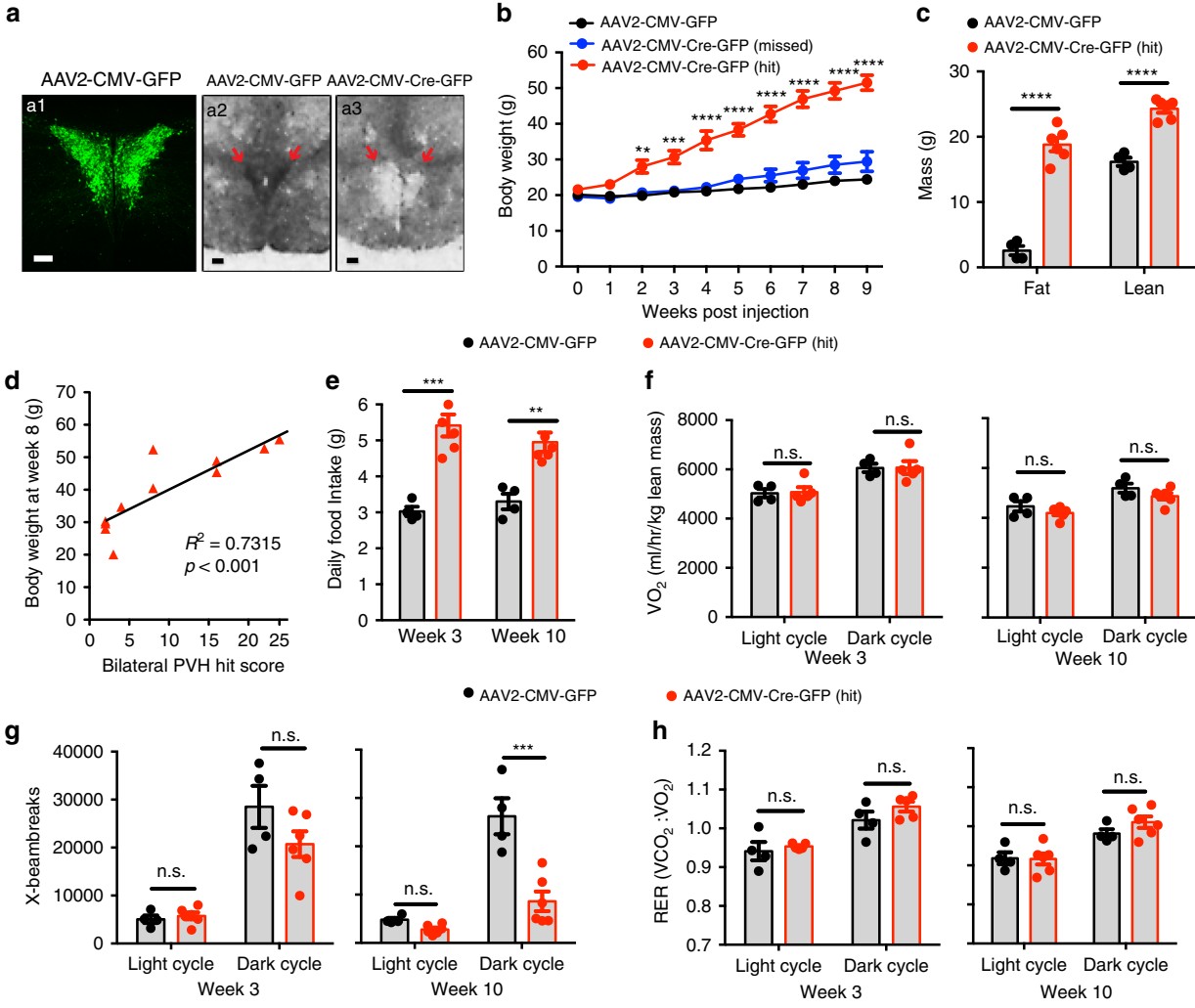

**Fig. 3 Deletion of the *Ntrk2* gene in the PVH of adult female mice leads to hyperphagic obesity. a** Bilateral injection of AAV2-CMV-GFP or AAV2-CMV-Cre-GFP (200 nl) into the PVH of *Ntrk2$^{lox/lox}$* mice. GFP fluorescence (a1) and *Ntrk2* mRNA in situ hybridization (a2, a3) are shown. Scale bars are 100 μm (a1) and 200 μm (a2, a3), respectively. **b** Body weight of *Ntrk2$^{lox/lox}$* mice injected with either AAV2-CMV-GFP or AAV2-CMV-Cre-GFP into the PVH bilaterally. $n = 4$ mice for AAV2-CMV-GFP, 5 mice for AAV2-CMV-Cre-GFP (missed), and 6 mice for AAV2-CMV-Cre-GFP (hit). Two-way ANOVA with post hoc Bonferroni multiple comparisons; $F_{(2, 108)} = 34.442$, $P < 0.0001$ for viral injection; **$P = 0.0025$ and ****$P < 0.0001$ for comparisons between AAV2-CMV-Cre-GFP (hit) and AAV2-CMV-GFP or AAV2-CMV-Cre-GFP (missed). **c** Body composition of *Ntrk2$^{lox/lox}$* mice 9 weeks after AAV injection. $n = 4$ mice for AAV2-CMV-GFP and 5 mice for AAV2-CMV-Cre-GFP (hit). Two-tailed unpaired *t* test; ****$P < 0.0001$. **d** Correlation between the extent of AAV2-CMV-Cre-GFP transduction in PVH and body weight at 8 weeks after injection. Data are from both AAV2-CMV-Cre-GFP (hit) and AAV2-CMV-Cre-GFP (missed) groups. PVH hit score is a sum of the extents of AAV transduction in 6 divisions of the PVH (see Supplementary Table 1 for detail). **e** Daily food intake of *Ntrk2$^{lox/lox}$* mice during week 3 and week 10 post injection. $n = 4$ mice for AAV2-CMV-GFP and five mice for AAV2-CMV-Cre-GFP (hit). Two-tailed unpaired *t* test; ** $P = 0.0058$ and *** $P = 0.0008$. **f** O$_2$ consumption of *Ntrk2$^{lox/lox}$* mice during week 3 and week 10 post injection. Two-tailed unpaired *t* test; n.s. not significant. **g** Locomotor activity of *Ntrk2$^{lox/lox}$* mice during week 3 and week 10 post injection. Two-tailed unpaired *t* test; n.s. not significant and **$P < 0.002$. **h** RER of *Ntrk2$^{lox/lox}$* mice during week 3 and week 10 post injection. Two-tailed unpaired *t* test; n.s. not significant. Animal numbers for **f**–**h** are 4 mice for AAV2-CMV-GFP and six mice for AAV2-CMV-Cre-GFP (hit). Error bars indicate SEM. Source data are provided as a Source Data file.

neurons are active to maintain physiological satiation during the light cycle. Interestingly, inhibition of PVH$^{TrkB}$ neurons also increased food intake during the dark cycle when mice are physiologically hungry and ingest the vast majority of their daily energy intake (Fig. 4d), suggesting that the anorexigenic PVH$^{TrkB}$ neurons are not fully silenced when mice are physiologically hungry. Conversely, CNO administration drastically reduced food intake of *Ntrk2$^{CreER/+}$* mice expressing hM3D(Gq)-mCherry during the dark cycle or after overnight fasting, compared with vehicle administration (Fig. 4e, f). The treatment induced Fos expression almost exclusively in hM3D(Gq)-mCherry-expressing PVH$^{TrkB}$ neurons (Supplementary Fig. 6).

These results indicate that activation of PVH$^{TrkB}$ neurons is sufficient to suppress appetite. Taken together, these chemogenetic experiments indicate that PVH$^{TrkB}$ neurons are partially silenced to allow for feeding when mice are hungry, while being active to maintain physiological satiation during the light cycle, and thus PVH$^{TrkB}$ neurons play a critical role in the regulation of appetite by promoting satiety.

**PVH$^{TrkB}$ neurons project to multiple brain areas.** We sought to identify the downstream sites through which PVH$^{TrkB}$ neurons regulate appetite. To map the projections of PVH$^{TrkB}$ neurons in the brain, we injected Cre-dependent AAV2-CAG-FLEX-tdTomato

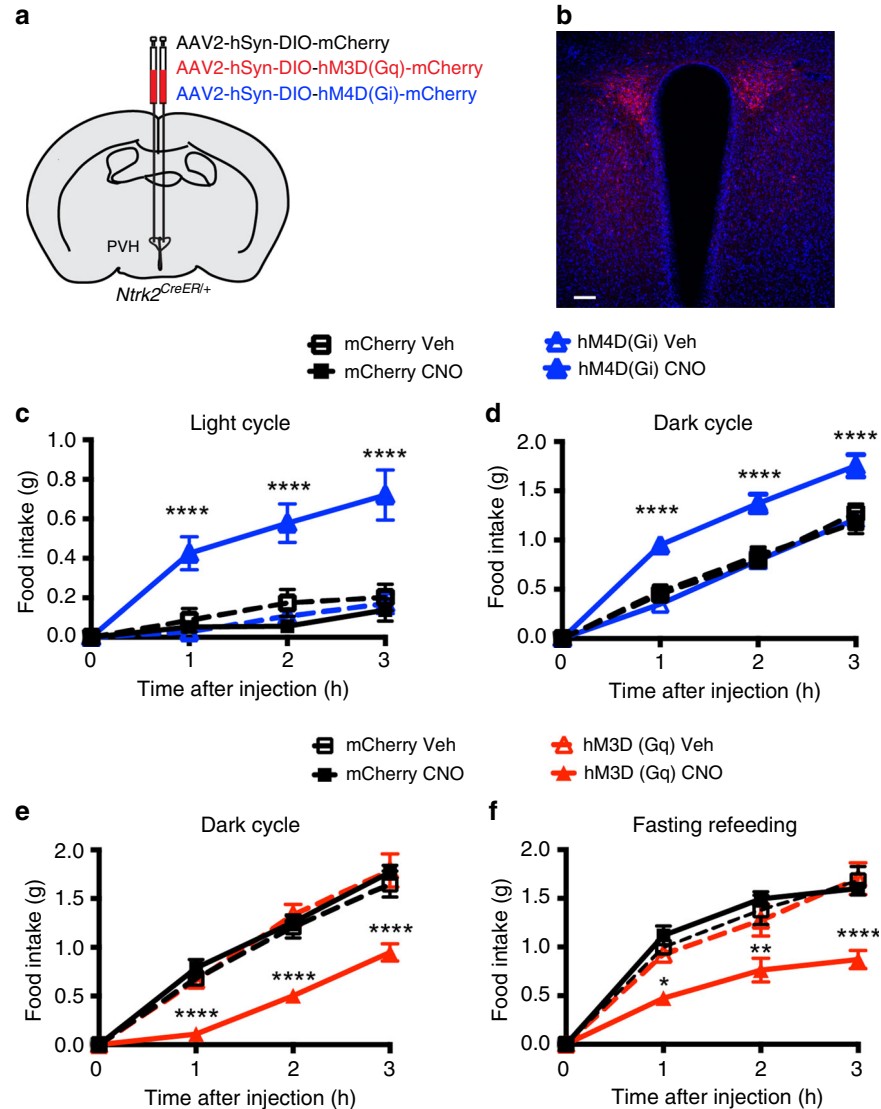

**Fig. 4 Chemogenetic modification of PVH$^{TrkB}$ neuronal activity alters food intake. a** A diagram showing bilateral injection of AAV2-hSyn-DIO-mCherry, AAV2-hSyn-DIO-hM3D(Gq)-mCherry, or AAV2-hSyn-DIO-hM4D(Gi)-mCherry into the PVH of $Ntrk2^{CreER/+}$ mice. **b** A confocal imaging showing mCherry expression of injected AAV in the PVH. Brain sections were counter-stained with DAPI. The scale bar represents 100 μm. **c** Inhibition of PVH$^{TrkB}$ neurons stimulated food intake during the light cycle. $n = 5$ and six mice for AAV2-hSyn-DIO-mCherry (mCherry) and AAV2-hSyn-DIO-hM4D(Gi)-mCherry [hM4D(Gi)] groups, respectively. Two-way ANOVA with post hoc Tukey's multiple comparisons; $F_{(3, 72)} = 35.23$, $P < 0.0001$ for virus and treatment; ****$p < 0.0001$ when compared to the hM4D(Gi)-vehicle (Veh) group. **d** Inhibition of PVH$^{TrkB}$ neurons stimulated food intake during the dark cycle. $n = 6$ and six mice for mCherry and hM4D(Gi) groups, respectively. Two-way ANOVA with post hoc Tukey's multiple comparisons; $F_{(3, 76)} = 31.49$, $P < 0.0001$ for virus and treatment; ****$p < 0.0001$ when compared to the hM4D(Gi)-Veh group. **e** Activation of PVH$^{TrkB}$ neurons suppressed food intake during the dark cycle. $n = 6$ and seven mice for AAV2-hSyn-DIO-mCherry (mCherry) and AAV2-hSyn-DIO-hM3D(Gq)-mCherry [hM3D(Gq)] groups, respectively. Two-way ANOVA with post hoc Bonferroni's multiple comparisons; $F_{(3, 22)} = 17.27$, $P < 0.0001$ for virus and treatment; ****$P < 0.0001$ when compared to the hM3D(Gq)-Veh group. **f** Activation of PVH$^{TrkB}$ neurons suppressed fasting-induced appetite. $n = 6$ and seven mice for mCherry and hM3D(Gq) groups, respectively. Two-way ANOVA with post hoc Bonferroni's multiple comparisons; $F_{(3, 22)} = 10.47$, $P < 0.0001$ for virus and treatment; *$P < 0.05$, **$P < 0.01$, and ****$P < 0.0001$ when compared to the hM3D(Gq)-Veh group. Error bars indicate SEM. Source data are provided as a Source Data file.

into the PVH of adult $Ntrk2^{CreER/+}$ mice (Fig. 5a1). After tamoxifen treatment, tdTomato nicely labeled PVH$^{TrkB}$ neurons across the rostral-caudal axis (Fig. 5a2–4). PVH$^{TrkB}$ neurons project to four main brain sites: the VMH (Fig. 5a5), the median eminence (ME; Fig. 5a5), the lateral parabrachial nucleus (LPBN; Fig. 5a6), and the nucleus tractus solitaris (NTS; Fig. 5a8). Lesser projections were detected in the medial parabrachial nucleus (MPBN; Fig. 5a6), the locus coeruleus (LC; Fig. 5a7), the dorsal motor nucleus of the vagus (DMV; Fig. 5a8), and the ventrolateral periaqueductal gray (vlPAG; Supplementary Fig. 7a). In the LPBN, axons of PVH$^{TrkB}$

neurons mainly terminate at the central compartment, which is next to the CGRP (calcitonin gene-related peptide)-marked external compartment[53] (Fig. 5b).

The projection to the ME is expected, as TrkB is expressed in a fraction of PVH neurons expressing GHRH (Supplementary Fig. 1e) and oxytocin (Fig. 1c), which project to the ME and the posterior pituitary via the ME, respectively, to release hormones. To validate other putative projection fields, we injected retrograde tracers, Alexa Fluor 488-conjugated cholera toxin subunit B (CTB488) or green retrobead (GRB), into each of these fields in

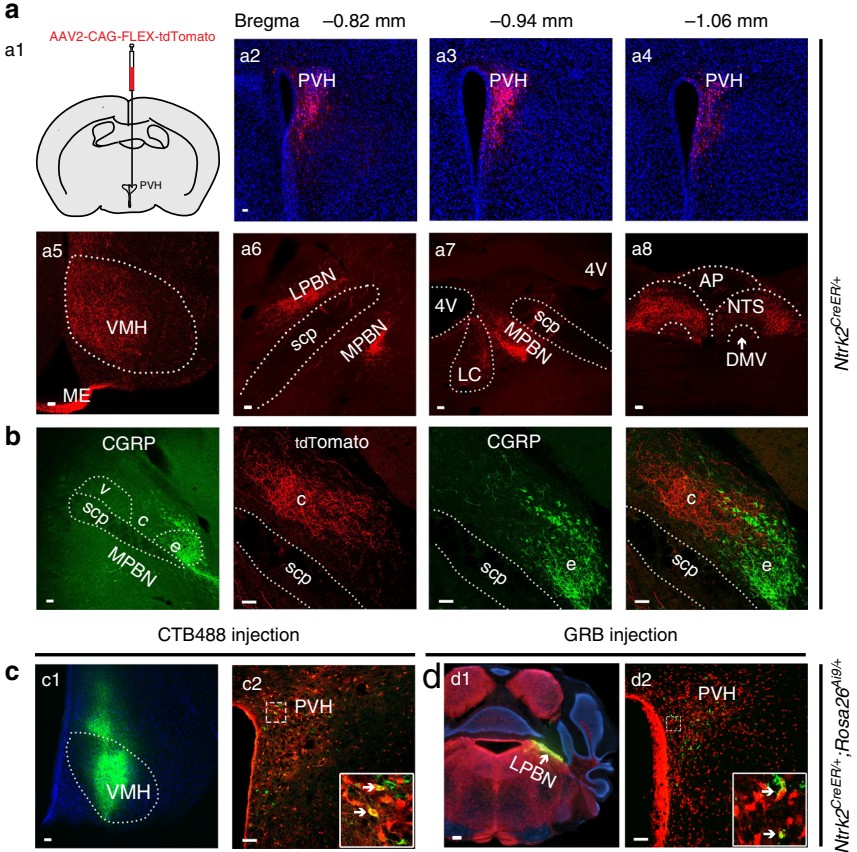

**Fig. 5 Projections of PVH^TrkB neurons. a** AAV2-CAG-FLEX-tdTomato was unilaterally injected into the PVH of *Ntrk2^CreER/+* mice (a1). After the mice were treated with tamoxifen, tdTomato-labeled PVH^TrkB neurons were present throughout the rostral-caudal axis (a2–a4). Axonal terminals of these neurons were found in the VMH, LPBN, MPBN, LC, NTS and DMV (a5–a8). **b** The LPBN includes the external (e), central (c), and ventral (v) compartments. Axonal terminals of PVH^TrkB neurons are mainly in the central compartment, which is next to the CGRP-marked external compartment. **c, d** Retrograde tracers, CTB488 (250 nl) or GRB (200 nl), were unilaterally injected into the VMH (**c**) or LPBN (**d**) of tamoxifen-treated *Ntrk2^CreER/+;Rosa26^Ai9/+* mice and labeled some tdTomato-expressing PVH^TrkB neurons. AP area postrema; DMV dorsal motor nucleus of the vagus; ME median eminence; LC locus coeruleus; LPBN lateral parabrachial nucleus; MPBN medial parabrachial nucleus; NTS nucleus tractus solitaris; PVH paraventricular hypothalamus; scp superior cerebellar peduncle; VMH ventromedial hypothalamus. Scale bars represent 200 µm in d1 and 50 µm in other panels. Source data are provided as a Source Data file.

tamoxifen-treated *Ntrk2^CreER/+;Rosa26^Ai9/+* mice. In a true projection field, retrograde tracers would be taken up at axonal terminals and retrogradely transported to the cell bodies of PVH^TrkB neurons. Indeed, we found that retrograde tracers injected into the VMH (Fig. 5c), LPBN (Fig. 5d), vlPAG, LC, and NTS (Supplementary Fig. 7) labelled some TrkB-expressing neurons in the PVH. These results show that PVH^TrkB neurons project to the central compartment of the LPBN (cLPBN), VMH, NTS, MPBN, DMV, vlPAG, ME, and pituitary.

**Projection-specific gene deletion.** In order to determine at which projection PVH TrkB is necessary for the regulation of appetite, we designed a two-virus system for projection-specific gene deletion. In this strategy, one viral vector is a retrograde virus [canine adenovirus 2 (CAV2)[54] or retrograde AAV (AAVretro)[55]] expressing yeast flippase (FLP), whereas the other viral vector is AAV2-Ef1a-fDIO-mCherry-P2A-Cre, in which expression of mCherry-P2A-Cre is dependent on FLP-mediated recombination at two sets of Frt sequences (Fig. 6a). If we stereotaxically inject FLP-expressing retrograde virus into the LPBN and AAV2-fDIO-mCherry-P2A-Cre into the PVH, the retrograde virus would be taken up at axonal terminals and retrogradely transported to the cell bodies of neurons that project to the LPBN. In the PVH, FLP expressed in neurons that project to the LPBN

would mediate recombination between the two sets of Frt sites in AAV2-Ef1a-fDIO-mCherry-P2A-Cre and invert the orientation of the mCherry-P2A-Cre sequence, leading to the expression of mCherry and Cre (Fig. 6a). If the two viral vectors are injected into *Ntrk2^lox/lox* mice, expressed Cre would delete the *Ntrk2* gene in PVH neurons that project to the LPBN.

We conducted a proof-of-principle experiment to test this projection-specific gene deletion approach in the well-defined substantia nigra → striatum projection. We injected CAV2-FLPo, which expresses optimized FLP, into the striatum and AAV2-Ef1a-fDIO-mCherry-P2A-Cre into the substantia nigra of *Bdnf^klox/+* mice (Fig. 6b1). *Bdnf^klox/+* mice contains a uniquely floxed *Bdnf* allele that starts to express β-galactosidase in BDNF-expressing cells when Cre recombines the allele[56]. We detected many mCherry-expressing neurons in the substantia nigra (Fig. 6b2), indicating that CAV2-FLPo was retrogradely transported to the substantia nigra from the striatum and induced mCherry expression from AAV2-Ef1a-fDIO-mCherry-P2A-Cre. Because mCherry and Cre are in the same cistron, mCherry-positive neurons should also express Cre, which would mediate recombination of the *Bdnf^klox* allele. Indeed, we found that some mCherry-positive neurons expressed β-galactosidase in the substantia nigra (Fig. 6b2–4), where ~50% dopaminergic neurons express BDNF[57]. As expected, we detected expression of neither mCherry nor β-galactosidase in the substantia nigra

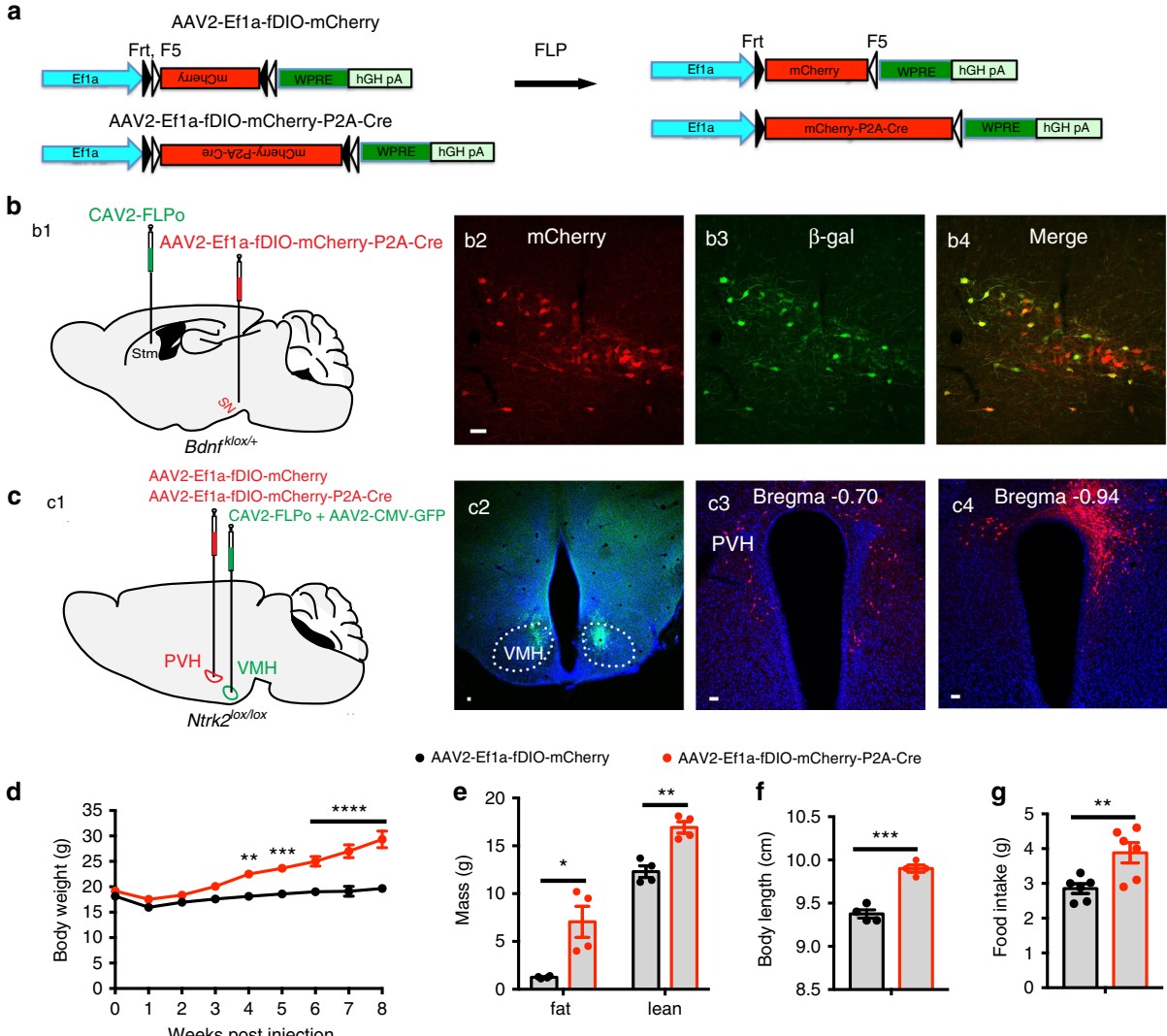

**Fig. 6 Deletion of the *Ntrk2* gene in PVH neurons projecting to the VMH leads to hyperphagia and obesity. a** Diagrams of AAV2-Ef1a-fDIO-mCherry and AAV2-Ef1a-fDIO-mCherry-P2A-Cre viral vectors for FLP-dependent expression of mCherry and Cre. **b** Deletion of the *Bdnf* gene in substantia nigra neurons projecting to the striatum. CAV2-FLPo (500 nl) and AAV2-Ef1a-fDIO-mCherry-P2A-Cre (200 nl) were stereotaxically injected into the striatum (Stm) and substantia nigra (SN) of *Bdnf^klox/+* mice, respectively (b1). Expression of mCherry and β-galactosidase (β-gal) was detected in the substantia nigra (b2-4). **c** Deletion of the *Ntrk2* gene in PVH neurons projecting to the VMH. A mixture of CAV2-FLPo + AAV2-CMV-GFP (350 nl, 5:1 ratio) and either AAV2-Ef1a-fDIO-mCherry (control) or AAV2-Ef1a-fDIO-mCherry-P2A-Cre (150 nl) were bilaterally injected into the VMH and PVH of *Ntrk2^lox/lox^* mice, respectively (c1). Co-injected AAV2-CMV-GFP was used to mark CAV2 injection sites (c2). Neurons that are transduced by both CAV2-FLPo and one of the two FLP-dependent AAV vectors would express mCherry (c3 and c4). **d–g** Body weight (**d**), fat mass and lean mass (**e**), body length (**f**), and daily food intake (**g**) of female *Ntrk2^lox/lox^* mice injected with CAV2-FLPo into the VMH and either AAV2-Ef1a-fDIO-mCherry or AAV2-Ef1a-fDIO-mCherry-P2A-Cre into the PVH. $n = 6$ mice for each group. Body weight data were analyzed using two-way ANOVA with Bonferroni *post hoc* tests; $F_{(1,\ 90)} = 124.0$, $P < 0.0001$ for virus. Other data were analyzed with two-tailed unpaired $t$ test; *$P = 0.012$ and **$P = 0.002$ for fat mass and lean mass, ***$P = 0.0002$ for body length, and ***$P = 0.0097$ for food intake. Error bars indicate SEM. Scale bars represent 50 µm. Source data are provided as a Source Data file.

when CAV2-FLPo was not injected into the striatum (Supplementary Fig. 8a). We further confirmed no leaky expression of mCherry-P2A-Cre from AAV2-Ef1a-fDIO-mCherry-P2A-Cre by injecting the virus into the PVH (Supplementary Fig. 8b). This experiment shows that our two-virus system works for projection-specific gene deletion.

**PVH TrkB regulates feeding through multiple projections.** We targeted the *Ntrk2* gene in PVH neurons that project to the NTS, VMH and LPBN, the three main projection fields of PVH^TrkB neurons. Retrograde virus CAV2-FLPe-GFP (expressing a fusion protein of GFP and the improved FLP), CAV2-FLPo, or AAV2retro-hSyn-FLPo-T2A-GFP were used to express FLP in

cell bodies of afferent neurons. We conducted this study only in female mice. Because *Ntrk2* deletion in the PVH produced more severe obesity in females than males (Fig. 3b and Supplementary Fig. 4h), we used female mice to more sensitively uncover the role of different PVH^TrkB projections in the control of appetite.

To delete the *Ntrk2* gene in PVH neurons projecting to the NTS (PVH^TrkB→NTS neurons), we injected CAV2-FLPe-GFP into the NTS and either AAV2-Ef1a-fDIO-mCherry (control) or AAV2-Ef1a-fDIO-mCherry-P2A-Cre into the PVH in *Ntrk2^lox/lox^* mice (Supplementary Fig. 8c). Because the NTS is a long structure, we infused a mixture of CAV2-FLPe-GFP and GRBs into two sites, the rostral part (Supplementary Fig. 8d1–3) and the caudal part (Supplementary Fig. 8d4–6) of the central NTS. Co-injected

GRBs allows us to determine if an injection hits the NTS (Supplementary Fig. 8d). In hit mice, we detected mCherry expression in the PVH (Supplementary Fig. 8e), indicating expression of Cre in AAV2-Ef1a-fDIO-mCherry-P2A-Cre injected mice, which would delete the $Ntrk2$ gene in positive neurons. We found comparable body weight (Supplementary Fig. 8f), body composition (Supplementary Fig. 8g) and food intake (Supplementary Fig. 8h) between mutant and control mice. These results suggest that TrkB in PVH$^{TrkB\rightarrow NTS}$ neurons is not essential for the control of food intake and body weight.

We next targeted the $Ntrk2$ gene in PVH neurons projecting to the VMH (PVH$^{TrkB\rightarrow VMH}$ neurons) by injecting a mixture of CAV2-FLPo and AAV2-CMV-GFP into the VMH and either AAV2-Ef1a-fDIO-mCherry or AAV2-Ef1a-fDIO-mCherry-P2A-Cre into the PVH in $Ntrk2^{lox/lox}$ mice (Fig. 6c1). Co-injected AAV2-CMV-GFP marks injection sites (Fig. 6c2). In our initial trial, we noticed lesions at CAV2-FLPo injection sites, likely due to an immune response to CAV2. Therefore, we treated $Ntrk2^{lox/lox}$ mice with the immunosuppressant cyclophosphamide (50 mg/kg, i.p.) once every 2 days from day −3 to day 13 as previously described[58]. Ten weeks after CAV2 injection, we did not detect any lesions in the VMH (Fig. 6c2), indicating that cyclophosphamide successfully blocked the immune response to CAV2. In mice with successful viral injection as evidenced by bilateral mCherry expression in the PVH (Fig. 6c3, c4), the Cre group had significantly higher body weight (Fig. 6d), fat mass and lean mass (Fig. 6e), body length (Fig. 6f), and food intake (Fig. 6g). These results show that deleting $Ntrk2$ in PVH$^{TrkB\rightarrow VMH}$ neurons leads to hyperphagia and obesity, indicating a crucial role for TrkB in PVH$^{TrkB\rightarrow VMH}$ neurons in appetite suppression.

Finally, we deleted the $Ntrk2$ gene in PVH neurons projecting to the LPBN (PVH$^{TrkB\rightarrow LPBN}$ neurons) by injecting AAV2retro-hSyn-FLPo-T2A-GFP into the LPBN and either AAV2-Ef1a-fDIO-mCherry or AAV2-Ef1a-fDIO-mCherry-P2A-Cre into the PVH in $Ntrk2^{lox/lox}$ mice (Fig. 7a1). We employed AAV2retro-hSyn-FLPo-T2A-GFP rather than CAV2-FLPo for the PVH → LPBN projection, because we found that AAVretro is more efficient in retrograde infection than CAV2 at this projection. AAV2retro-hSyn-FLPo-T2A-GFP expresses GFP at injection sites (Fig. 7a2), allowing us to assess the accuracy of each injection. In mice with successful viral injection as evidenced by GFP expression in the LPBN (Fig. 7a2) and mCherry expression in the PVH (Fig. 7a3, a4), the Cre group had significantly higher body weight (Fig. 7b), fat mass and lean mass (Fig. 7c), body length (Fig. 7d), and food intake (Fig. 7e). These results show that deleting $Ntrk2$ in PVH$^{TrkB\rightarrow LPBN}$ neurons also leads to hyperphagia and obesity, indicating a crucial role for TrkB in PVH$^{TrkB\rightarrow LPBN}$ neurons in the control of food intake.

**VMH- and LPBN-projecting PVH$^{TrkB}$ neurons are distinct.** Because the TrkB receptor at both PVH$^{TrkB}$ projections to the VMH and the LPBN is crucial for the control of appetite, we next investigated the possibility that the same subset of PVH$^{TrkB}$ neurons send collaterals to the VMH and the LPBN. We stereotaxically injected retrograde tracers CTB488 and Alexa Fluor 594-conjugated CTB (CTB594) into the VMH and the LPBN, respectively (Fig. 7f1–3). We found that CTB594 mainly labeled cell bodies in the rostral and medial parts of the PVH, whereas CTB488 largely in the medial and caudal parts of the PVH (Fig. 7f4–7). No neurons were detected to contain both CTB488 and CTB594 (Fig. 7f4–7). These results indicate that two distinct subtypes of PVH$^{TrkB}$ neurons project to the VMH and the LPBN. We validated this finding by examining axonal terminals of mCherry-labeled PVH neurons that project to the LPBN in mice where AAV2-Ef1a-fDIO-mCherry and AAV2retro-hSyn-FLPo-

T2A-GFP were injected into the PVH and the LPBN, respectively (Supplementary Fig. 9a–d). We detected axonal terminals of these neurons in the LPBN, but not in the VMH (Supplementary Fig. 9e–h).

The observation that abolishment of $Ntrk2$ expression in PVH$^{TrkB\rightarrow VMH}$ and PVH$^{TrkB\rightarrow LPBN}$ neurons leads to hyperphagic obesity suggests that these two subtypes of neurons respond to changes in nutritional state. We tested this hypothesis by examining Fos induction in these two subsets of neurons in response to refeeding. We injected CTB488 into either the VMH (Fig. 8a) or the LPBN (Fig. 8d) of tamoxifen-treated $Ntrk2^{CreER/+}$; $Rosa26^{Ai9/+}$ mice. After the mice were extensively handled, they were fasted overnight and a half of them were refed for 2 h. We found that more CTB488-labeled PVH$^{TrkB}$ neurons expressed Fos in refed mice than in fasted mice (Fig. 8b, c, e, f), indicating that refeeding activates PVH$^{TrkB\rightarrow VMH}$ and PVH$^{TrkB\rightarrow LPBN}$ neurons. These results together with the findings from $Ntrk2$ deletion and manipulation of neuronal activity, shows that feeding activates PVH$^{TrkB\rightarrow VMH}$ and PVH$^{TrkB\rightarrow LPBN}$ neurons to suppress appetite.

**BDNF increases firing of PVH$^{TrkB}$ neurons.** BDNF has been shown to potentiate neurotransmission at excitatory synapses[19]. As the vast majority of PVH neurons are excitatory[18,19] and activation of PVH$^{TrkB}$ neurons suppress food intake (Fig. 4e, f), we tested the hypothesis that BDNF increases the activity of PVH$^{TrkB}$ neurons to suppress appetite. We injected AAV2-Ef1a-fDIO-mCherry and AAV2retro-Ef1a-DIO-FLPo into the PVH and the LPBN of $Ntrk2^{CreER/+}$ mice, respectively, to mark PVH$^{TrkB\rightarrow LPBN}$ neurons with mCherry (Fig. 9a). Whole-cell patch-clamp recordings of these labeled neurons (Fig. 9b, c) revealed that bath application of BDNF (25 ng/ml) increased neuronal firing with a latency of 15–40 min (Fig. 9d, e).

**Discussion**

This study has identified the PVH as a main site where TrkB signaling works to suppress food intake. Using a projection-specific gene deletion strategy, we have further narrowed down the action site of TrkB signaling to two distinct subtypes of PVH neurons that project to the LPBN and the VMH, respectively. We have found that refeeding activates these two subsets of PVH$^{TrkB}$ neurons, BDNF application increases the firing rate of PVH$^{TrkB\rightarrow LPBN}$ neurons, and chemogenetic activation of PVH$^{TrkB}$ neurons inhibits food intake during the dark cycle or after overnight food deprivation. Thus, our study indicates that signals reflecting repletion induce BDNF release, which in turn increases the activity of PVH$^{TrkB}$ neurons to promote satiety through their outputs to the LPBN and the VMH.

We found that deletion of the $Ntrk2$ gene with the Sim1-Cre transgene led to hyperphagia and reduced energy expenditure in mice. This observation is in agreement with the previous finding that BDNF expressed in the PVH suppresses food intake and promotes adaptive thermogenesis[22]. Collectively, these results demonstrate that BDNF-TrkB signaling plays a critical role in the control of both energy intake and energy expenditure. When we selectively deleted the $Ntrk2$ gene in the PVH using stereotaxic injection of Cre-expressing AAV, mutant mice displayed marked hyperphagia but normal energy expenditure. It is likely that TrkB signaling in other brain areas that express Sim1-Cre is involved in the regulation of energy expenditure. One of these brain areas could be the DMH, in which $Ntrk2$ deletion significantly reduces energy expenditure[28]. Injection of BDNF protein into the PVH was reported to increase energy expenditure[39]. This could be a result of activation of the TrkB receptor at axonal terminals of

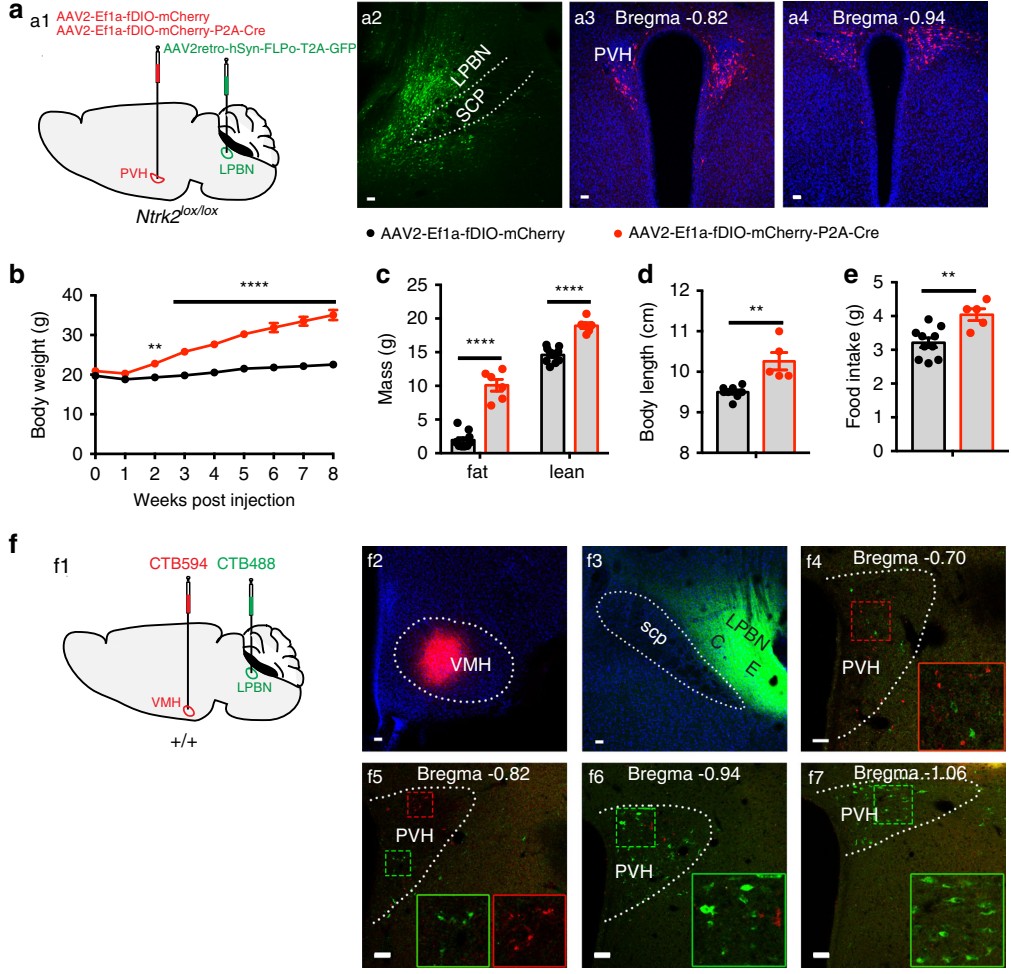

**Fig. 7 Deletion of the *Ntrk2* gene in PVH neurons projecting to the LPBN leads to hyperphagia and obesity. a** Deletion of the *Ntrk2* gene in PVH neurons projecting to the LPBN. AAV2retro-hSyn-FLPo-T2A-GFP (200 nl) and either AAV2-Ef1a-fDIO-mCherry (control) or AAV2-Ef1a-fDIO-mCherry-P2A-Cre (150 nl) were bilaterally injected into the LPBN and PVH of *Ntrk2*lox/lox mice, respectively (a1). GFP marks injection sites of AAV2retro-hSyn-FLPo-T2A-GFP (a2). Neurons that are transduced by both AAV2retro-hSyn-FLPo-T2A-GFP and one of the two FLP-dependent AAV vectors would express mCherry (a3 and a4). **b–e** Body weight (**b**), fat mass and lean mass (**c**), body length (**d**), and food intake (**e**) of female *Ntrk2*lox/lox mice injected with AAV2retro-hSyn-FLPo-T2A-GFP into the LPBN and either AAV2-Ef1a-fDIO-mCherry or AAV2-Ef1a-fDIO-mCherry-P2A-Cre into the PVH. *n* = 10 mice in the control group and 6 mice in the Cre group. Body weight data were analyzed using two-way ANOVA with Bonferroni *post hoc* tests; $F_{(1, 126)} = 488.7$, $P < 0.0001$ for virus. Other data were analyzed with two-tailed unpaired *t* test; ****$P < 0.0001$ for fat mass and lean mass, **$P = 0.0014$ for body length, and **$P = 0.004$ for food intake. **f** Retrogradely labeling of PVH neurons. CTB594 (250 nl) and CTB488 (200 nl) were injected into the VMH and the LPBN, respectively (f1-3). Some retrogradely labeled neurons are in the PVH (f4-7). Error bars indicate SEM. Scale bars represent 50 µm. Source data are provided as a Source Data file.

neurons that project to the PVH or a non-specific effect of recombinant BDNF.

A genetically defined group of neurons in a specific brain structure usually have multiple projection fields, and the powerful optogenetic approach is commonly employed to uncover the role of the neurons at each projection in animal behaviors. However, this approach is not suitable for uncovering gene function and is not ideal for examination of the long-term effect of altering neuronal activity. The projection-specific gene deletion described here makes it possible to determine the function of a gene in a specific neurocircuit. If deletion of the gene also alters the function of the targeted neurocircuit, this experiment will also reveal both short-term and long-term impacts of altering the activity of the circuit on behaviors and physiology.

Using the projection-specific gene deletion approach, we have identified two subtypes of PVH^TrkB neurons (PVH^TrkB→LPBN neurons and PVH^TrkB→VMH neurons) in which TrkB signaling is necessary to suppress appetite. Deletion of the *Ntrk2* gene in

either of these two subtypes of neurons led to hyperphagia and obesity. Conversely, we found that deletion of the *Ntrk2* gene in PVH^TrkB→NTS neurons did not affect energy balance. It is unlikely that this negative outcome is due to inadequate *Ntrk2* deletion in NTS-projecting PVH neurons, as we showed that even unilateral *Ntrk2* deletion in the PVH led to obesity. We have found that refeeding activates both PVH^TrkB→LPBN and PVH^TrkB→VMH subsets of neurons and that chemogenetic stimulation/inhibition of PVH^TrkB neurons greatly decreases/increases food intake, respectively. These results suggest that abolishment of TrkB signaling impairs the function of these two subtypes of neurons, which leads to deficits in transmitting satiation signals to their target neurons in the LPBN and VMH. These results also indicate that the PVH^TrkB→LPBN and PVH^TrkB→VMH neurocircuits are satiating and that BDNF is a key modulator of the two anorexigenic neurocircuits. As BDNF potently potentiates synaptic transmission[19], it is likely that TrkB signaling enhances synaptic transmission in the two subtypes of

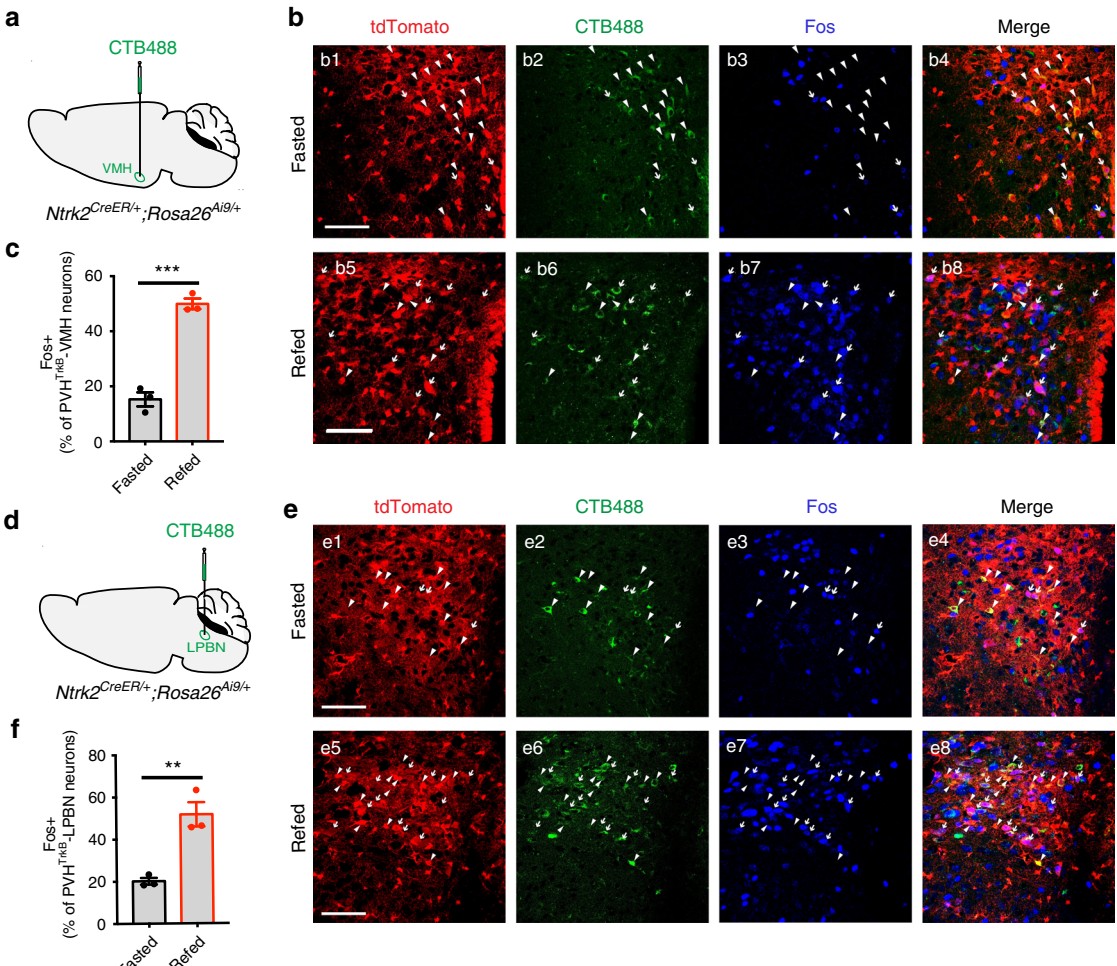

**Fig. 8 Refeeding activates PVH$^{TrkB}$ neurons projecting to the VMH and the LPBN. a–c** Refeeding after overnight food deprivation induced Fos expression in PVH$^{TrkB}$ neurons projecting to the VMH. PVH$^{TrkB\rightarrow VMH}$ neurons were labeled by tdTomato and CTB488 injected into the VMH in tamoxifen-treated *Ntrk2$^{CreER/+}$;Rosa26$^{Ai9/+}$* mice (**a**). Arrowheads denote neurons positive for tdTomato and CTB488, while arrows indicate neurons positive for tdTomato, CTB488 and Fos (**b**). Fos induction in PVH$^{TrkB\rightarrow VMH}$ neurons under fasting and refeeding conditions was quantified (**c**). n = 3 mice in the fasted group and three mice in the refed group. Two-tailed unpaired *t* test; ***$P = 0.0004$. **d–f** Refeeding after overnight food deprivation induced Fos expression in PVH$^{TrkB}$ neurons projecting to the LPBN. PVH$^{TrkB\rightarrow LPBN}$ neurons were labeled by tdTomato and CTB488 injected into the LPBN in tamoxifen-treated *Ntrk2$^{CreER/+}$; Rosa26$^{Ai9/+}$* mice (**d**). Arrowheads denote neurons positive for tdTomato and CTB488, while arrows indicate neurons positive for tdTomato, CTB488 and Fos (**e**). Fos induction in PVH$^{TrkB\rightarrow LPBN}$ neurons under fasting and refeeding conditions was quantified (**f**). $n = 3$ mice in the fasted group and three mice in the refed group. Two-tailed unpaired *t* test; **$P = 0.006$. Error bars indicate SEM. Scale bars represent 50 μm. Source data are provided as a Source Data file.

PVH$^{TrkB}$ neurons. Indeed, we found that BDNF application increased the firing rate of PVH$^{TrkB\rightarrow LPBN}$ neurons. It would be important to understand which neurons release BDNF to facilitate activation of PVH$^{TrkB}$ neurons in response to refeeding, what are the targets of PVH$^{TrkB}$ neurons in the LPBN and VMH, and how TrkB signaling modulates synaptic function in PVH$^{TrkB\rightarrow LPBN}$ and PVH$^{TrkB\rightarrow VMH}$ neurons in future studies.

The vast majority of PVH neurons are glutamatergic neurons[59]. These neurons have been shown to provide excitatory drive to the AgRP neurons in the arcuate nucleus[60], the cLPBN[26], and the pre-locus coeruleus (pLC)[27] to regulate satiety. This study identifies one more neurocircuit out of the PVH, i.e. PVH$^{TrkB\rightarrow VMH}$, to regulate satiety.

Although ablation of the oxytocin receptor-expressing neurons in the hindbrain leads to overeating[45], ablation of PVH oxytocin neurons in adult mice has no effect on body weight, food intake, or energy expenditure[50]. Consistent with this observation, we found that deletion of the *Ntrk2* gene in PVH$^{OXT}$ neurons did not affect energy balance. Thus, PVH$^{TrkB\rightarrow LPBN}$ and PVH$^{TrkB\rightarrow VMH}$

neurons should be distinct from oxytocin neurons. Previous work shows that LPBN-projecting PVH$^{MC4R}$ (PVH$^{MC4R\rightarrow LPBN}$) neurons and PVH$^{BDNF}$ neurons also promote satiety[22,26], although it remains unknown where PVH$^{BDNF}$ neurons project to suppress appetite. As we detected low co-localization of TrkB with either MC4R or BDNF, PVH$^{TrkB\rightarrow LPBN}$ and PVH$^{TrkB\rightarrow VMH}$ neurons could be distinct from PVH$^{MC4R\rightarrow LPBN}$ and PVH$^{BDNF}$ neurons. A recent study shows that PDYN-expressing pLC-projecting PVH (PVH$^{PDYN\rightarrow pLC}$) neurons regulate satiety and are distinct from PVH$^{MC4R}$ neurons[27]. If PVH$^{BDNF}$ neurons are distinct from PVH$^{PDYN\rightarrow pLC}$ neurons, at least 5 subtypes of PVH neurons (PVH$^{MC4R\rightarrow LPBN}$, PVH$^{BDNF}$, PVH$^{PDYN\rightarrow pLC}$, PVH$^{TrkB\rightarrow LPBN}$, and PVH$^{TrkB\rightarrow VMH}$) have been identified to promote satiety. It would be important to investigate how these neurons coordinately regulate satiety and body weight. This is especially true for PVH$^{MC4R\rightarrow LPBN}$ and PVH$^{TrkB\rightarrow LPBN}$ neurons, which project to the same target field, the cLPBN. It would be intriguing to know whether these two subtypes of neurons innervate the same neurons in the cLPBN and receive inputs from the same afferent neurons.

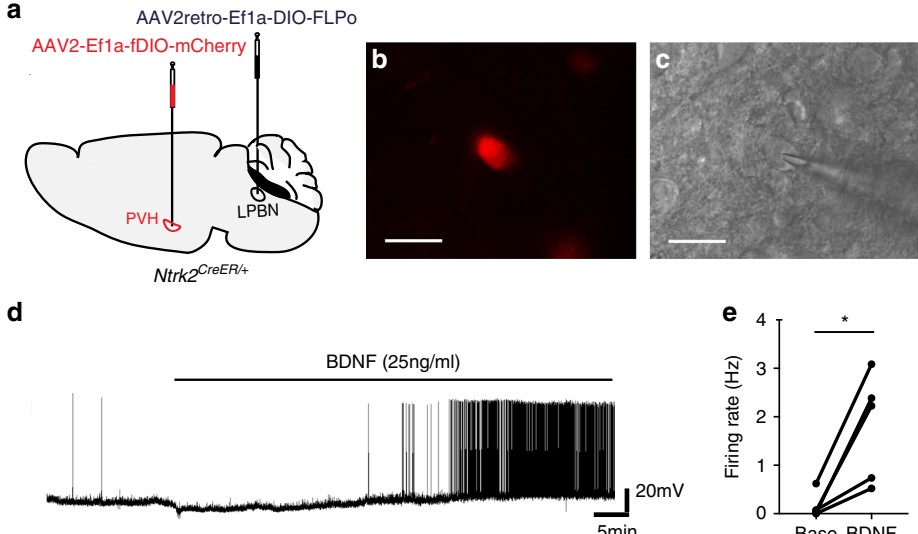

**Fig. 9 BDNF increases the firing rate of PVH$^{TrkB}$ neurons projecting to the LPBN. a** PVH$^{TrkB\rightarrow LPBN}$ neurons were labeled with injection of AAV2-Ef1a-fDIO-mCherry into the PVH and AAV2retro-Ef1a-DIO-FLPo into the LPBN of $Ntrk2^{CreER/+}$ mice. **b, c** Patch-clamp recording of an mCherry-labeled PVH$^{TrkB\rightarrow LPBN}$ neuron. Scale bars represent 25 μm. **d** A representative trace showing neuronal firing before and after bath application of BDNF. **e** Quantification of baseline neuronal firing (Base) and neuronal firing 40 min after bath application of BDNF (BDNF). Two-tailed paired Student's $t$ test; *$P = 0.019$. Source data are provided as a Source Data file.

These future studies will provide insights into the organization and function of the neural network that regulates appetite.

## Methods

**Animals.** Floxed $Ntrk2$ ($Ntrk2^{lox}$; also known as TrkB$^{lox}$)[43], floxed $Ntrk2$-LacZ ($Ntrk2^{fBZ}$; also known as fBZ)[42], and floxed $Bdnf$ ($Bdnf^{klox}$)[56] mouse strains were previously described. $Ntrk2^{lox}$ mice were backcrossed to C57BL/6J mice for at least 15 generations before they were used in this study. Sim1-Cre (stock No: 006395), MC4R-tau-GFP (stock No: 008323), Rosa26$^{Ai9}$ [Gt9(ROSA)26Sor$^{tm99CAG-tdTomato)Hze}$/J; stock No: 007909] and C57BL6/J (stock No: 000664) mouse strains were obtained from the Jackson Laboratory. The $Ntrk2^{CreER/+}$ (also known as TrkB$^{CreER}$) mouse strain[32] was kindly provided by Dr. David Ginty at Harvard Medical School. We generated $Ntrk2^{CreER/+}$;Rosa26$^{Ai9/+}$ mice by crossing $Ntrk2^{CreER/+}$ mice to Rosa26$^{Ai9/+}$ mice. To induce nuclear localization of the CreER fusion protein in mice harboring the $Ntrk2^{CreER}$ allele, we treated the mice with tamoxifen (2 mg per animal, i.p.) once a day for 7 consecutive days in 6–8-week-old mice. Animals were used for experiments 7 days after the last tamoxifen injection. $Ntrk2^{lox/lox}$ mice were crossed with Sim1-Cre;$Ntrk2^{lox/lox}$ mice to generate Sim1-Cre;$Ntrk2^{lox/lox}$ mutant mice and $Ntrk2^{lox/lox}$ control mice. All mice used in this study were maintained in the C56BL6/J genetic background. Mice were maintained on a 12-h/12-h light/dark cycle with *ad libitum* access to water and regular laboratory chow (Teklad Rodent Diet 2019; 3.3 kcal/g energy density, 22% kcal from fat) unless otherwise specified. All experiments were performed in accordance with relevant guidelines and regulations regarding the use of experimental animals. The Animal Care and Use Committees at The Scripps Research Institute Florida approved all animal procedures used in this study.

**Immunohistochemistry.** For neuropeptide immunohistochemistry, we injected $Ntrk2^{CreER/+}$;Rosa26$^{Ai9/+}$ mice with colchicine (20 μg per animal, i.c.v.; Tocris) to impair protein transportation in neuronal processes. Forty-eight hours after colchicine injection, mice were deeply anaesthetized with avertin and transcardially perfused with phosphate buffered saline (PBS), followed by 4% paraformaldehyde in PBS. Coronal brain sections (40 μm in thickness) were obtained using a sliding microtome (Leica SM2000R). Brain sections were rinsed once with Tris buffered saline (TBS; 10 mM Tris-HCl, 150 mM NaCl, pH7.5), incubated with blocking buffer (0.4% Triton X-100, 1% bovine serum albumin, and 10% goat serum or horse serum in TBS) for 1 h at room temperature, and then incubated with primary antibodies diluted in the blocking buffer overnight at room temperature. The following primary antibodies were used: chicken anti-β-galactosidase (1:3,000; abcam #ab9361), rabbit anti-NeuN (1:1000; Millipore #ABN78), rabbit anti-GFAP (1:400; Sigma #G9269), mouse anti-oxytocin (1:5000; Millipore #MAB5296), rabbit anti-CRH (1:500; Millipore #AB1760), mouse anti-GFP (1:1000; Clontech #632460), rabbit anti-somatostatin (1:500; immunostar #20067), rabbit anti-GHRH (1:500; Immunostar #22938), rabbit anti-vasopressin (1:5000; Millipore #AB1565), mouse anti-tyrosine hydroxylase (1:10,000; Sigma #T1299), rabbit anti-c-Fos antibody (1:5000; abcam #ab208942), rabbit anti-prodynorphin (1:200; abcam #ab11137), rabbit anti-DsRed (1:1000; TaKaRa #632496), and rabbit anti-TRH (1:10,000; generous gift from Dr. Martin Wessendorf, University of Minnesota). After three washes in TBS, sections were incubated with appropriate fluorescent secondary antibodies (1:500; Jackson ImmunoResearch) for 1 h at room temperature. Sections were washed three times in TBS, mounted onto slides, and coverslipped with a mounting medium containing 4′,6-diamidino-2-phenylindole (DAPI; Vector Laboratories). Fluorescent images were captured using a Nikon C2+ confocal microscope or a Leica TCS SP8 microscope.

**Plasmids and viruses.** For generation of pAAV-Ef1a-fDIO-mCherry and pAAV-Ef1a-fDIO-mCherry-P2A-Cre, pAAV-Ef1a-fDIO-EYFP (a gift from Karl Deisseroth; Addgene plasmid #55641) was used as backbone. pAAV-Ef1a-fDIO-EYFP was digested with AscI and NheI site to remove EYFP. The sequence encoding mCherry and mCherry-P2A-Cre were obtained by PCR using plasmid pLM-CMV-R-Cre (a gift from Michel Sadelain; Addgene plasmid #27546) as template. PCR products were digested with AscI and NheI and cloned into the backbone of pAAV-Ef1a-fDIO-EYFP to generate plasmid pAAV-Ef1a-fDIO-mCherry and pAAV-Ef1a-fDIO-mCherry-P2A-Cre. pAAV-hSyn-FLPo-T2A-GFP was generated using pAAV-hSynapsin-Flpo (a gift from Massimo Scanziani; Addgene plasmid #60663) as backbone. We replaced the FLPo sequence containing a stop codon with a FLPo without a stop codon using BamHI and HindIII sites to generate an intermediate plasmid. The T2A-GFP sequence was amplified by PCR using plasmid PX461 (a gift from Feng Zhang; Addgene plasmid #48140) as template and inserted into the intermediate plasmid at HindIII site to generate pAAV-hSyn-FLPo-T2A-GFP. AAV2-Ef1a-fDIO-mCherry, AAV2-Ef1a-fDIO-mCherry-P2A-Cre and AAV2retro-hSyn-Flpo-T2A-GFP were packaged by UNC Vector Core. Plasmid AAV pEF1a-DIO-FLPo-WPRE-hGHpA was obtained from Addgene (#87306; a gift of Li Zhang), and AAV2retro-DIO-FLPo virus was packaged by ViGene Biosciences Inc. AAV2-CMV-GFP, AAV2-CMV-GFP-Cre and AAV2-CAG-FLEX-tdTomato viruses were purchased from UNC vector Core. AAV2-hSyn-DIO-mCherry, AAV2-hSyn-DIO-hM3Dq-mCherry and AAV2-hSyn-DIO-hM4Di-mCherry were purchased from Addgene.

To construct pCAV2-FLPe-GFP, we inserted the Flpe-GFP sequence into the CAV shuttle plasmid[61] to generate pShuttle-FLPe-GFP. The FLPe-GFP sequence was isolated from pCAG-Flpe-GFP (a gift from Connie Cepko, Addgene plasmid #13788). The AscI-PacI fragment of the shuttle construct was purified and used to generate pCAV2-FLPe-GFP through homologous recombination in E. coli as described[62]. To generate pCAV2-FLPo, we began with pCAV-FLEx$^{loxP}$-Flp[63]. pCAV-FLEx$^{loxP}$-Flp contains the CAV-2 genome deleted in the E1 and E3 regions, and a cassette containing a CMV promoter, two loxP sites, an inverted FLPo open reading frame, and an SV40 polyA signal. pCAV-FLEx$^{loxP}$-Flp was incubated with recombinant Cre recombinase (New England Biolabs) and then transformed into DHα bacteria. A clone containing the FLPo cassette in the 5′ → 3′ orientation was identified by restriction digestion analyses and denoted pCAV2-FLPo. Viral vectors CAV2-FLPe-GFP and CAV2-FLPo were generated, amplified, and purified as previously described[61].

**Stereotaxic injection of virus and CTB**. Mice were deeply anesthetized with isoflurane and were placed onto a stereotaxic apparatus (David Kopf, Model 940). A small incision was made to expose the skull, and a small hole was drilled on skull above the injection site. A Nanofil 33-gauge needle (World Precision Instruments, #NF33BV-2) was inserted into the brain and the virus was injected at a rate of 25 nl per min using Microsyringe Pump (World Precision Instruments, Micro4). After infusion, the needle was stayed in the same position for 5 min. Following injections, mice received subcutaneous injection of Loxicom (0.5 mg per kg) to relieve pain. Stereotaxic coordinates from the bregma and skull used for this study were PVH (AP: −0.5 mm, ML: ± 0.25 mm, DV: −5.4 mm), VMH (AP: −1.4 mm, ML: ± 0.4 mm, DV: −5.95 mm), LPBN (AP: −5 mm, ML: ± 1.5 mm, DV: −3.8 mm), vlPAG (AP: −4.72 mm, ML: ± 0.68 mm, DV: −2.70 mm), and LC (AP: −5.4 mm, ML: ± 0.91 mm, DV: −4.1 mm). Stereotaxic coordinates for the NTS relative to the calamus scriptorius were rostral NTS (AP: + 0.85 mm, ML: ± 0.58 mm, DV: −0.42 mm) and caudal NTS (AP: −0.00 mm, ML: ± 0.26 mm, DV: −0.4 mm). After physiological measurements are completed, animals were perfused with saline and 4% paraformaldehyde and their brains were sliced for examination of injection sites. Mice with missed injections were either assigned to a separate group or excluded in final data analyses. Cholera toxin b subunit conjugated with Alexa dye 488 (CTB-488; Life Technologies, # C-34775) or 594 (CTB-594; Life Technologies, # C-34777) at 1 mg per ml was used for retrograde tracing.

**Physiological measurements**. In the *Ntrk2* gene deletion experiments, body weight was measured weekly. Body length (naso-anal) was measured from anesthetized mice in the end of each experiment. For food intake, mice were individually housed, and daily food intake was measured over 5 days after 3 days of acclimation. Body composition was determined using a Minispec LF-50/mq 7.5 NMR Analyzer (Brucker Optics). $VO_2$ and locomotor activity were assessed with a comprehensive lab animal monitoring system (CLAMS, Columbus Instruments). Ambulatory activity was measured as light beam breaks in the XY horizontal plane.

**Chemogenetic manipulation of food intake**. Mice injected with AAV2-hSyn-DIO-mCherry, AAV2-hSyn-DIO-hM3D(Gq)-mCherry or AAV2-hSyn-DIO-hM4D(Gi)-mCherry were singly housed for at least 2 weeks with daily handling prior to experiment. For food intake measurement during light cycle, mice were transferred to a new cage without food at 7:30 am and injected intraperitoneally with vehicle or CNO (Sigma-Aldrich #C08325) at 8:30 a.m. Pre-weighed food was then added to each cage at 9:00 a.m. and food intake was measured hourly for 3 h. For food intake measurement during dark cycle, mice were transferred to a new cage without food at 3 p.m. and injected with vehicle or CNO at 5:30 pm. Pre-weighed food was then added to each cage at 6:00 p.m. and food intake was measured hourly from 7:00 p.m. to 9:00 p.m. For food intake measurement after overnight fasting, mice were transferred to a new cage with alpha-dri bedding without food at 6:00 p.m. for overnight fasting. On the following day, mice were injected with vehicle or CNO at 9:30 a.m. and pre-weighed food was added to each cage at 10:00 a.m. and food intake was measured hourly for 3 h. The CNO dosage was 1 mg per kg body weight for mice injected with AAV2-hSyn-DIO-hM3D(Gq)-mCherry and 3 mg per kg body weight for mice injected with AAV2-hSyn-DIO-hM4D(Gi)-mCherry.

**In situ hybridization**. Freshly dissected mouse brains were quickly frozen in a 2-methylbutane (Fisher Scientific #O3551-4) dry ice bath. Subsequently, 20-µm thick cryostat sections were collected on Superfrost Plus slides (Fisherbrand #12–550–15) and used for in situ hybridization. Radioactive in situ hybridization of brain sections was performed using $^{35}$S-labeled riboprobes as previously described[11]. To generate antisense riboprobes, the mouse cDNA sequence for *Ntrk2* (GenBank accession number X17647, nucleotides 188–722) was amplified by PCR and cloned into pBluscript II KS (-) plasmid (Stratagene). Antisense probes were synthesized with T3 RNA polymerase (Promega #2083) and labeled with alpha-$^{35}$S-CTP (PerkinElmer #NEG064H250UC) and alpha-$^{35}$S-UTP (PerkinElmer #NEG039H250UC). Brain sections were postfixed in 4% paraformaldehyde for 1 h and acetylated for 10 min with 0.1 M triethanolamine hydrochloride/0.25% acetic anhydride (pH 8.0). After dehydration, hybridization was performed with $^{35}$S labeled probes at 55 °C for overnight. After hybridization, sections were treated with RNase A (200 µg/ml, Fisher Scientific # BP2539100) to remove unannealed probes. After dehydration, sections were exposed to Kodak BioMax MR films (Carestream Health) for 3 days. Developed film was scanned at 1200 dpi for image processing. Fluorescence in situ hybridization was performed using the RNAscope Multiplex Fluorescent Reagent Kit v2 (#323100, Advanced Cell diagnostics). Brain sections were fixed with pre-chilled 10% formalin at 4 °C for 15 min, and dehydrated with 50%, 75%, and 100% ethanol. Sections were air-dried for 20 min, pretreated with protease IV for 30 min, washed 2 times in PBS and incubated with the target probes (Ntrk2-C3; #423611-C3, tdTomato-C2; #317041-C2, Glp1r-C2; #418851) for 2 h at 40 °C. The signal was amplified by subsequent incubation with v2Amp1 for 30 min, v2Amp2 for 30 min, v2Amp3 for 15 min, and followed by TSA Plus Fluorescein/Cyanine5 (NEL754001KT, Akoya Biosciences) for 30 min at 40 °C. Slides were counterstained with DAPI and images were acquired using a Nikon C2+ confocal microscope.

**Quantification of co-expression**. For fluorescent in situ hybridization, brains were sectioned at 14-µm thickness, and ten series of PVH-containing sections were collected (one slide for each series and 4–5 sections on each slide). One slide was used for each in situ hybridization. Sections on each slide approximately covers the PVH from Bregma −0.58 to −1.06 mm. In situ hybridization signals were examined for co-expression on each cell marked with DAPI staining in the PVH. For fluorescent immunohistochemistry, every other brain section (40-µm thick) from each *Ntrk2^{CreER/+};Rosa26^{Ai9/+}* brain (6–7 sections in total) was used. Immunoreactive PVH neurons in all these sections were examined for co-expression with tdTomato. DAPI staining was used to determine the PVH structure. TrkB colocalization with oxytocin was examined in brain sections from two mice, while TrkB colocalization with other PVH markers was examined in brain sections from one mouse.

**Quantitative real-time PCR**. Mouse brain was rapidly extracted and placed in ice-cold 1x HBSS. Brain slice from bregma −0.34 to bregma −1.22 was obtained using a vibratome (Leica VT 1200s, Germany), and the PVH was punched out bilaterally using a 20 gauge blunt-end needle. The punched brain tissues were transferred from needle into 1.5-ml tubes. Total RNA was isolated from PVH using PureLink RNA mini kit combined with Trizol reagent (Life Technologies, Carlsbad, CA). To remove genomic DNA contamination, the RNA samples were treated with RNase-free DNase. First strand cDNA was reverse transcribed using 50 ng of total RNA and Super-Script™ III (Invitrogen, Carlsbad, CA) according to the manufacture's protocol. Quantitative PCR was performed on step-one thermal cycler (Applied Biosystems, Foster City, CA) with SYBR® Green PCR master mix (Roche). Levels for *Ntrk2* mRNA encoding the full-length TrkB (forward, gtgctgatggcagagggta; reverse, tattttcaccagcaggttctct), *Mc4r* mRNA (forward, cccggacggaggatgctat; reverse, tcgccacgatcactagaatgt),) were normalized over 18s rRNA (forward, ccgcagctggaataatgga; reverse, ccctcttaatcatggcctca) mRNA in the same sample. As the *Mc4r* gene does not contain an intron, a parallel experiment using no reverse transcriptase control was used to monitor genomic DNA contamination.

**Slice whole-cell electrophysiology**. Male mice at 12–16 weeks of were decapitated under isoflurane anesthesia. Brains were rapidly removed and placed in ice-cold sucrose cutting solution containing (mM): 72 sucrose, 83 NaCl, 2.5 KCl, 1 CaCl$_2$, 5 MgCl$_2$, 1 NaH$_2$PO$_4$, 22 D-glucose, and 26.0 NaHCO$_3$ saturated with 95% O$_2$/5% CO$_2$. Coronal brain slices containing the PVH were obtained using a vibratome (Leica VT 1200s, Germany), and then transferred to artificial cerebrospinal fluid (aCSF) composed of (mM): 126 NaCl, 2.5 KCl, 21.4 NaHCO$_3$, 1.2 NaH$_2$PO$_4$, 1.2 MgCl, 2.4 CaCl$_2$ and 10 D-glucose, equilibrated with 95% O$_2$ and 5% CO$_2$ at 32 °C for 30 min. Brain slices were maintained and recorded at room temperature (22–25 °C). Then, the brain slices were gently transferred to a recording chamber (RC-27, Warner Instruments, Hamden, CT), and the chamber was perfused with circulated oxygenated aCSF at a flow rate of 2–3 ml per min. PVH TrkB neurons projecting to LPBN were visually identified in slices using an infrared-differential interference contrast microscope and Texas Red fluorescent filter (Scientifica, UK). Whole-cell patch-clamp recordings were performed using borosilicate glass pipettes (ID: 0.68 mm, OD: 1.2 mm, WPI, Sarasota, FL) of 3–5 MΩ pulled with a micropipette puller (P-1000; Sutter Instrument, Novato, CA). The intracellular solution for current clamp recording contained (mM): 128 K-gluconate, 10 KCl, 10 HEPES, 1 EGTA, 1 MgCl$_2$, 5 Na$_2$-ATP, 0.3 Mg-GTP, 0.3 CaCl$_2$, adjusted to pH 7.3 with KOH. To test the effect of BDNF on the activity of PVH$^{TrkB}$ neurons projecting to LPBN, current clamp recordings were performed in continuous mode with membrane potential held between −55 and −60 mV to minimize the spontaneous firing rate[64]. Baseline activities were recorded for at least 20 min, then BDNF (Millipore Corp, Billerica, MA) was added to the circulating aCSF to reach a final concentration of 25 ng/ml. Signals were acquired with Multiclamp700B and Digidata 1550A (Molecular Devices, San Jose, CA). Data were low-pass filtered at 2.9 KHz and sampled at 10 kHz. The neuronal firing rate was analyzed with Clampfit 10.6 software. Data were expressed as average firing rate within 20 min just before and 40 min post BDNF treatment.

**Statistical analysis**. Statistical analyses were performed using Prism 7 (GraphPad Software). All data are expressed as mean ± SEM. Data were tested using two-tailed unpaired or paired *t* test or two-way ANOVA with post hoc Bonferroni's or Tukey's multiple comparisons for statistical significance. Experiments in this study were repeated independently for two to three times. No statistical methods were used to determine sample size for each experiment. The animal numbers were estimated based on our previous studies. Half of the brain sections from each virus-injected mouse were used for histological examination of injection sites. Mice with weak or off-target viral expression were excluded from final data analysis. Each micrograph shows a representative situation for each group of mice.

**Reporting summary**. Further information on research design is available in the Nature Research Reporting Summary linked to this article.

## Data availability

The data that support the findings of this study are available from the corresponding author upon reasonable request. The source data underlying Figs. 1–9 and Supplementary Figs. 1, 3–5, and 8 are provided as a Source Data file.

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

## Acknowledgements

We thank David Ginty for the *Ntrk2^CreER* mouse strain, Martin Wessendorf for the anti-TRH antibody, and Jessica Houtz and Shaw-wen Wu for critical reading of this paper. This work was supported by US National Institutes of Health grants to BX (R01 DK103335 and R01 DK105954).

## Author contributions

J.J.A. and B.X. conceived the study. J.J.A., C.E.K., and B.X. designed the study. J.J.A. did colocalization immunohistochemistry and analyzed Sim1-Cre;*Ntrk2^lox/lox* conditional knockout mice. C.E.K., J.J.A., and G.Y.L. did stereotaxic injection of AAV-Cre-GFP. C.E. K. and J.J.A. analyzed PVH-specific *Ntrk2* mutant mice. J.J.A. constructed viral vectors for projection-specific gene deletion, did DREADD experiments, and examined Fos induction after refeeding. J.J.A. constructed pCAV2-FLPe-GFP. E.J.K. produced CAV2-FLPe-GFP and CAV2-FLPo. J.J.A. and C.E.K. did projection tracing and projection-specific *Ntrk2* deletion. J.W.T performed electrophysiological recordings. J.J. A. and B.X. prepared figures. B.X. and J.J.A. wrote the paper with input from all authors.

## Competing interests

The authors declare no competing interests.
