## [Peer Review File · Nature Communications]

Reviewers' Comments:

Reviewer #1:

Remarks to the Author:

In this research article by An et al., the authors report the role of TrkB receptor-expressing neurons in the PVH in suppressing feeding behavior. By crossing Ntrk2CreER mice with Rosa26Ai9 mice, the authors first examined expression of TrkB within the PVN, and reported only minor overlap with other neural populations expressing TRH, PDYN, somatostatin, GHRH, CRH, tyrosine hydroxylase or vasopressin, though the overlap with oxytocin is higher, at 18%. The authors concluded that these PVHTrkB neurons are previously unidentified and went on to investigate the role of TrkB in circuit regulation using cre-dependent deletion of TrkB by either crossing Sim1-Cre mice with Ntrk2lox/lox mice or injecting Cre AAVs into the PVH of Ntrk2lox/lox mice. Both manipulations led to obesity and hyperphagia. In addition, chemogenetic stimulation/inhibition of PVHTrkB neurons reduces/increases food intake, respectively, which is consistent with view that BDNF-TrkB signaling potentiates synaptic transmission. The authors further went on to map out the projections of PVHTrkB neurons and identified VMH and LPBN as two key downstream pathway of appetite suppression. By combining retrograde FLP and cre, the authors were able to delete TrkB expression in PVHTrkB neurons that specifically project to the VMH/LPBN, both of which led to hyperphagia and obesity. Lastly, LPBN projecting PVHTrkB neurons are shown to increase Fos expression during a refeeding paradigm. Overall, this study is comprehensive and nicely demonstrates the importance of TrkB receptor expression in regulation of feeding neurocircuits, mostly by way of circuit-specific gene deletion. The data are generally convincing, and the results will be of interest to the field. I only have a few comments that might help improving the overall impact of the study.

Major Concerns:

The authors showed in Fig 1 the expression of TrkB within PVH at 3 rostral-caudal positions, but lacks a more thorough characterization of the expression within the full PVH. The PVH extends between bregma -0.22 mm and -1.22 mm, therefore examining the expression of TrkB on a broader scale would provide more information. After examining colocalization of TrkB with various other markers, the authors concluded that 'the majority of PVHTrkB neurons are distinct from previously defined neuronal populations'. These analyses rely on specific antibodies, and it is a little bit puzzling that the left panel of tdTomato expressing neurons show different patterns, densities as well as resolutions. It would probably be more convincing to include complementary methodologies such as RNAscope colocalization analyses (see Li C, et al. Cell Metab, 2019, PMID:30472090). Furthermore, include the population of GLP-1r will be of great help, since GLP-1r expressing neurons are important for regulation of feeding behavior as well. Moreover, it would be more rigorous to conduct quantifications and present the data in pie charts or some other plots. Detailed quantifications of methods should be included.

It is mentioned in the text that BDNF-TrkB signaling is known to potentiate synaptic transmission in the hippocampus. In order to prove this to be the case in PVH, a brief electrophysiological study in slice would be helpful to truly confirm that TrkB activation by BDNF indeed promotes synaptic transmission and/or neural excitability. Or at least include the possible functional impact of TrkB deletion on the overall excitability as well as the impact of the TrkB signaling on synaptic transmission. Do we know where are the sources of BDNF release in the PVH in the context of feeding behavior?

Minor Concerns:

-The statement regarding Fig. 2g-i 'These results indicate that TrkB signaling in Sim1-Cre cells promote physical activity and energy expenditure' lacks evidence, as this could be secondary to the effect on body weight (at 6 weeks female mice already show increased body weight in the KO group). Some discussion will help address this issue.

-In the discussion section, 'Deletion of the Ntrk2 gene in one of these two subtypes of neurons led to hyperphagia and obesity', please change 'one' to 'either'.

Reviewer #2:

Remarks to the Author:

This manuscript examines TrkB-containing neurons in the PVN, their role in the regulation of energy balance and critical downstream targets. This cell population is distinct from BDNF containing neurons and cells containing other known metabolic markers including MC4R. Overall, experiments are well executed and powered. The findings move the field forward, considering that little is known regarding the circuitry mediating the robust effects of BDNF-TrkB signaling on energy balance control. They define two novel circuits involving PVN TrkB+ neurons projecting to the VMH or LPBN that critically regulate metabolic function. Addressing the following points would further strengthen this manuscript.

1. Immunolabeling studies defining the peptidergic phenotype of VMH and LPBN neurons targeted by PVN TrkB+ cells would further inform this newly identified circuits.
2. Was TrkB deleted from the SCN in mutants generated using Sim-Cre? It looks like it from Fig. 2B. Considering that dysregulation of circadian rhythms can impact feeding, this needs to be considered in the overall conclusions.
3. Basal metabolic rate was decreased in TrkB Sim-Cre mutants but not in adult TrkB mutants generated by AAV-cre delivery to the PVN of floxed TrkB mice? This discrepancy is not reconciled by the authors.

Reviewer #3:

Remarks to the Author:

An and colleagues examine the role of TrkB receptor (Ntrk2) expression in the PVH in the regulation of energy balance. Using a combination of Cre-mediated gene deletion and chemogenetic manipulation of TrkB receptor PVH neurons, the authors demonstrate that PVHTrkB neurons play an important role in bodyweight maintenance predominantly through the regulation of feeding. PVHTrkB neurons show little overlap with other PVH populations thought to play a role in energy balance regulation. PVHTrkB neurons project to several brain regions involved in energy balance control and anterograde tracing approaches are supplemented with retrograde labeling. The authors use a novel combination of retrograde viral vectors to delete Ntrk2 from subsets of PVHTrkB neurons based on their projection pathway. Specifically the authors argue that Ntrk2 expression in PVHTrkB ->LPBN and PVHTrkB ->VMN pathways are critical for bodyweight maintenance, whereas Ntrk2 in PVHTrkB ->NTS projecting neurons has a minimal role in bodyweight control. The innovation and novelty of the paper's findings come from genetic manipulation of subsets of PVHTrkB neurons based on their projection targets. The effect of deletion of Ntrk2 from the PVH is impressive, and is on the scale of that seen with loss of melanocortin-4 receptor action in the PVH. The authors are to be complimented on the design and execution of their experiments as well as the clarity of the manuscript. There is, however, limited insight to the mechanism through which loss of Ntrk2 affects PVH neuron function which somewhat limits the novelty of the findings.

In addition, I do have several concerns regarding the data presented:

1. Interpretation of the co-expression of Ntrk2 with other markers in the PVH (Figure 1) is

dependent on the fidelity of Ntrk2-Cre activity following tamoxifen treatment as representative of endogenous Ntrk2 activity. Colocalization of Cre (or Td-tomato) mRNA with endogenous Ntrk2 mRNA expression by in situ hybridization is needed to quantify the validity of the inducible Cre expression in the PVH. This is highlighted by the apparent discrepancy of the number of Ntrk2 PVH cells seen with viral injection (Fig 5) vs that seen using the Rosa26 Tomato reporter. This is important because the phenotype elicited upon SimCre deletion of Ntrk2 deletion mimics the loss of Mc4R action in the PVH in terms of body weight, hyperphagia and body length (a rather unique feature of Mc4R-associated obesity).

2. SimCre deletion of Ntrk2 results in profound obesity and increased body length. As BDNF-TrkB signaling is involved in neural development and synaptic function and the obesity phenotype produced by loss of Ntrk2 from the PVH is very similar to Mc4R knockout, is Mc4R expression in the PVH altered following Ntrk2 loss? This is an important extension from the colocalization data presented in Figure 1.

3. The innovation and novelty of the paper lies in the attempt to manipulate PVHTrkB neurons based on their projection targets using several retrogradely transported Flp-recombinase viruses in combination with Flp-dependent Cre expression. Using this approach the authors present data demonstrating that Ntrk2 in PVHTrkB ->VMN and PVHTrkB ->LPBN microcircuits is required for bodweight control and normal feeding and the effect size is fairly equivalent (~10-12gm change over 8weeks post-injection). In both cases, however, an important control group is not described: Ntrk2 lox/lox mice injected with the flp-dependent Cre virus ALONE in the PVH. A control experiment is shown in Supplemental Figure 6 but this is in the striatum not the PVH; indeed some Cre activity appears to be present, as bGal staining (green cells) is detected in the image presented. Any independent Cre expression from this virus would recombine the Ntrk2 lox/lox gene in the PVH and mimic the experiments conducted in Figure 3.

4. Given the critical importance of the non-overlapping microcircuitry between the PVH->VMN and PVH->PBN to a central theme of the manuscript, additional support for the independent nature of these circuits would be beneficial. For example, when retroAAV-Flp-2aGFP is injected into the PBN, presumably it is transported to the PVH cell body to generate Flp recombinase and GFP. Expression of GFP should be an adequate tracing tool for these neurons and will allow the investigators to determine if PVHTrkB neurons labeled at the terminals in the PBN also project to the VMN.

5. The paper would also be strengthened by combining a PVHTrkB->PBN and PVHTrkB->VMN manipulation to determine whether these two projection pathways effectively recapitulate the obesity phenotype seen with pan-PVH deletion of Ntrk2. Based on the data presented, each projection subset contributes 10-12gm of excess bodyweight which is quite close to the total body mass gained by panPVH deletion (~25gm; figure 3)

6. Given the novelty of the PVH to VMN satiety circuit, it is unclear why the investigators chose to demonstrate Fos changes associated with feeding only in the PVH->PBN circuit (Fig 7g,h).

7. The fact that unilateral deletion of Ntrk2 also leads to obesity should be addressed in more detail. This is especially relevant given the fact that the loss of significant Ntrk2 from oxytocin neurons or from PVH->NTS projecting neurons has little effect on energy balance.

Minor concerns:

In Figure 2, the description of Ntrk2^{BZ/+} allele is confusing; I realize that it is referenced, but the authors would aid the reader in providing a description of the allele. The legend states that Figure 2A shows Ntrk2 deletion; however, in this figure bGal expression is a proxy for Ntrk2 expression. Also, representation of SimCre mediated Ntrk2 deletion in Fig 2B,C should show close ups of the hypothalamus in regions SimCre is expected to be active in order to allow the reader to more clearly visualize the deletion of Ntrk2.

For most chemogenetic activation experiments, Fos expression is following administration of CNO is used to demonstrate that transduced neurons are activated by CNO. This may be a useful addition for this manuscript as a means to determine if PVH neurons neighboring those transduced by the 3Dq-DREADD are also being activated indirectly in response to CNO.

In Figure 3d, a "hit score" is represented graphically, but never defined. What is a "hit score"? Also representative images of the PVH (Fig 3a2) should be provided in increased magnification, similar to that represented by the fluorescent image in Fig3a1.

The "proof of concept" experiment using CAV2-FlpO into the striatum and Flp-dependent mCherry-2Acre could be moved to supplemental data as its lengthy description detracts from the flow of the paper.

We thank the three reviewers for their overall positive assessments and constructive comments. We have addressed reviewers' concerns by including new data and revising the text in this revised manuscript. All changes in the main text can be tracked in the submitted word file. We believe that these revisions have greatly strengthened the manuscript. Here are our point-by-point response to reviewers' comments.

Reviewer #1

In this research article by An et al., the authors report the role of TrkB receptor-expressing neurons in the PVH in suppressing feeding behavior. By crossing *Ntrk2CreER* mice with *Rosa26Ai9* mice, the authors first examined expression of TrkB within the PVN, and reported only minor overlap with other neural populations expressing TRH, PDYN, somatostatin, GHRH, CRH, tyrosine hydroxylase or vasopressin, though the overlap with oxytocin is higher, at 18%. The authors concluded that these PVHTrkB neurons are previously unidentified and went on to investigate the role of TrkB in circuit regulation using cre-dependent deletion of TrkB by either crossing *Sim1-Cre* mice with *Ntrk2lox/lox* mice or injecting Cre AAVs into the PVH of *Ntrk2lox/lox* mice. Both manipulations led to obesity and hyperphagia. In addition, chemogenetic stimulation/inhibition of PVHTrkB neurons reduces/increases food intake, respectively, which is consistent with view that BDNF-TrkB signaling potentiates synaptic transmission. The authors further went on to map out the projections of PVHTrkB neurons and identified VMH and LPBN as two key downstream pathway of appetite suppression. By combining retrograde FLP and cre, the authors were able to delete TrkB expression in PVHTrkB neurons that specifically project to the VMH/LPBN, both of which led to hyperphagia and obesity. Lastly, LPBN projecting PVHTrkB neurons are shown to increase Fos expression during a refeeding paradigm. Overall, this study is comprehensive and nicely demonstrates the importance of TrkB receptor expression in regulation of feeding neurocircuits, mostly by way of circuit-specific gene deletion. The data are generally convincing, and the results will be of interest to the field. I only have a few comments that might help improving the overall impact of the study.

Major Concerns:

The authors showed in Fig 1 the expression of TrkB within PVH at 3 rostral-caudal positions, but lacks a more thorough characterization of the expression within the full PVH. The PVH extends between bregma -0.22 mm and -1.22 mm, therefore examining the expression of TrkB on a broader scale would provide more information. After examining colocalization of TrkB with various other markers, the authors concluded that 'the majority of PVHTrkB neurons are distinct from previously defined neuronal populations'. These analyses rely on specific antibodies, and it is a little bit puzzling that the left panel of tdTomato expressing neurons show different patterns, densities as well as resolutions. It would probably be more convincing to include complementary methodologies such as RNAscope colocalization analyses (see Li C, et al. *Cell Metab*, 2019, PMID:30472090). Furthermore, include the population of GLP-1r will be of great help, since GLP-1r expressing neurons are important for regulation of feeding behavior as well. Moreover, it would be more rigorous to conduct quantifications and present the data in pie charts or some other plots. Detailed quantifications of methods should be included.

We examined TrkB expression throughout the PVH in every other brain section. In Fig. 1a, we use images at 3 rostral-caudal positions to show that TrkB is expressed throughout the PVH. We have added pie charts to present co-expression data.

Because different PVH markers are expressed in distinct PVH divisions, the left panels of representative tdTomato-expressing neurons are from the PVH at different rostral-caudal positions. Thus, the patterns of tdTomato-expression neurons in these images look different. We have replaced the left panels with the same magnification of images.

We have used RNAscope to show that tdTomato expression in tamoxifen-treated *Ntrk2*^{CreER/+}; *Rosa26*^{Ai9/+} mice is a good indicator of TrkB expression (Fig. 1a). We also used the technique to examine co-expression of *Ntrk2* mRNA with *Glp1r* mRNA because GLP1R antibodies do not work for immunohistochemistry in our hands. We realized one issue when we used in situ hybridization for examination of TrkB expression neurons. *Ntrk2* mRNA is in neurons and astrocytes as well as in dendrites due to its long 3' untranslated region, which makes it challenging to examine co-expression of *Ntrk2* mRNA with another mRNA. Therefore, we stuck with the tdTomato reporter in the vast majority of co-expression experiments.

We have added the detail of quantification methods to this revised manuscript.

It is mentioned in the text that BDNF-TrkB signaling is known to potentiate synaptic transmission in the hippocampus. In order to prove this to be the case in PVH, a brief electrophysiological study in slice would be helpful to truly confirm that TrkB activation by BDNF indeed promotes synaptic transmission and/or neural excitability. Or at least include the possible functional impact of TrkB deletion on the overall excitability as well as the impact of the TrkB signaling on synaptic transmission. Do we know where are the sources of BDNF release in the PVH in the context of feeding behavior?

We did some recordings on PVH^{TrkB} neurons that project to the LPBN. We found that BDNF application increased firing of action potentials, indicating that BDNF-TrkB signaling promotes excitability of these neurons. The new data are shown in new Figure 9.

We do not know where are the sources of BDNF release in the PVH in the context of feeding behavior.

Minor Concerns:

-The statement regarding Fig. 2g-i 'These results indicate that TrkB signaling in Sim1-Cre cells promote physical activity and energy expenditure' lacks evidence, as this could be secondary to the effect on body weight (at 6 weeks female mice already show increased body weight in the KO group). Some discussion will help address this issue.

We have added some discussion on energy expenditure in this revised manuscript.

-In the discussion section, 'Deletion of the *Ntrk2* gene in one of these two subtypes of neurons led to hyperphagia and obesity', please change 'one' to 'either'.

Corrected.

Reviewer #2

This manuscript examines TrkB-containing neurons in the PVN, their role in the regulation of energy balance and critical downstream targets. This cell population is distinct from BDNF containing neurons and cells containing other known metabolic markers including MC4R. Overall, experiments are well executed and powered. The findings move the field forward, considering that little is known regarding the circuitry mediating the robust effects of BDNF-TrkB signaling on energy balance control. They define two novel circuits involving PVN TrkB+ neurons projecting to the VMH or LPBN that critically regulate metabolic function. Addressing the following points would further strengthen this manuscript.

1. Immunolabeling studies defining the peptidergic phenotype of VMH and LPBN neurons targeted by PVN TrkB+ cells would further inform this newly identified circuits.

As neurons in the VMH and LPBN have not been carefully characterized yet, it will take a large amount of efforts to define the peptidergic phenotype of VMH and LPBN neurons targeted by PVH^{TrkB}

neurons. Although this is an important and interesting topic, we think that it is out of the scope of this manuscript.

2. Was TrkB deleted from the SCN in mutants generated using Sim-Cre? It looks like it from Fig. 2B. Considering that dysregulation of circadian rhythms can impact feeding, this needs to be considered in the overall conclusions.

The *Ntrk2* gene is not deleted in the SCN of Sim1-Cre;*Ntrk2*^{lox/lox} mice. The image for the mutant does not have the SCN in the previous Fig. 2b. We have revised Fig. 2b by using two more comparable images.

3. Basal metabolic rate was decreased in TrkB Sim-Cre mutants but not in adult TrkB mutants generated by AAV-cre delivery to the PVN of floxed TrkB mice? This discrepancy is not reconciled by the authors.

We think that basal metabolic rate was not decreased in Sim1-Cre;*Ntrk2*^{lox/lox} mice, because oxygen consumption rate in the light cycle was comparable in control and mutant mice (Fig. 2g, h). Energy expenditure was decreased in Sim1-Cre;*Ntrk2*^{lox/lox} mice, but not in adult PVH-specific *Ntrk2* mutant mice. We think that the discrepancy is due to *Ntrk2* deletion in brain regions other than the PVH in Sim1-Cre;*Ntrk2*^{lox/lox} mice (Supplementary Fig. 3a-f). We address this issue in the second paragraph of the Discussion section.

Reviewer #3 (Remarks to the Author):

An and colleagues examine the role of TrkB receptor (*Ntrk2*) expression in the PVH in the regulation of energy balance. Using a combination of Cre-mediated gene deletion and chemogenetic manipulation of TrkB receptor PVH neurons, the authors demonstrate that PVHTrkB neurons play an important role in bodyweight maintenance predominantly through the regulation of feeding. PVHTrkB neurons show little overlap with other PVH populations thought to play a role in energy balance regulation. PVHTrkB neurons project to several brain regions involved in energy balance control and anterograde tracing approaches are supplemented with retrograde labeling. The authors use a novel combination of retrograde viral vectors to delete *Ntrk2* from subsets of PVHTrkB neurons based on their projection pathway. Specifically the authors argue that *Ntrk2* expression in PVHTrkB →LPBN and PVHTrkB →VMN pathways are critical for bodyweight maintenance, whereas *Ntrk2* in PVHTrkB →NTS projecting neurons has a minimal role in bodyweight control. The innovation and novelty of the paper's findings come from genetic manipulation of subsets of PVHTrkB neurons based on their projection targets. The effect of deletion of *Ntrk2* from the PVH is impressive and is on the scale of that seen with loss of melanocortin-4 receptor action in the PVH. The authors are to be complimented on the design and execution of their experiments as well as the clarity of the manuscript. There is, however, limited insight to the mechanism through which loss of *Ntrk2* affects PVH neuron function which somewhat limits the novelty of the findings.

In addition, I do have several concerns regarding the data presented:

1. Interpretation of the co-expression of *Ntrk2* with other markers in the PVH (Figure 1) is dependent on the fidelity of *Ntrk2*-Cre activity following tamoxifen treatment as representative of endogenous *Ntrk2* activity. Colocalization of Cre (or Td-tomato) mRNA with endogenous *Ntrk2* mRNA expression by in situ hybridization is needed to quantify the validity of the inducible Cre expression in the PVH. This is highlighted by the apparent discrepancy of the number of *Ntrk2* PVH cells seen with viral injection (Fig 5) vs that seen using the Rosa26 Tomato reporter. This is important because the phenotype elicited upon SimCre deletion of *Ntrk2* deletion mimics the loss of Mc4R action in the PVH

in terms of body weight, hyperphagia and body length (a rather unique feature of Mc4R-associated obesity).

We have performed in situ hybridization for tdTomato mRNA and *Ntrk2* mRNA in tamoxifen-treated *Ntrk2^{CreER/+}; Rosa26^{Ai9/+}* mice. The new data show that tdTomato expression efficiently and faithfully indicates TrkB expression (Fig. 1a). When we carried out the neuronal projection experiment (Fig. 5), we intentionally injected a limited number of AAV2-CAG-FLEX-tdTomato viruses in order to restrict viral transduction within the PVH, so that not all TrkB neurons would express tdTomato in this case.

In general, obese mice have an increased body length with exception of ob/ob and db/db mice. For example, mice homozygous for the mutation at the STAT3 binding site of the leptin receptor have an increased body length (Bates et al., 2003, Nature 421: 856-859). *Ntrk2* deletion in the dorsomedial hypothalamus also increases body length (Liao et al., 2019, PNAS 116: 3256-3261). Thus, obesity in association with increased body length may not necessarily mean disruption of MC4R signaling.

2. SimCre deletion of *Ntrk2* results in profound obesity and increased body length. As BDNF-TrkB signaling is involved in neural development and synaptic function and the obesity phenotype produced by loss of *Ntrk2* from the PVH is very similar to Mc4R knockout, is Mc4R expression in the PVH altered following *Ntrk2* loss? This is an important extension from the colocalization data presented in Figure 1.

We injected AAV-Cre-GFP or AAV-GFP into *Ntrk2^{lox/lox}* mice to generate a new batch of PVH-specific *Ntrk2* mutant or control mice. We did not detect a significant reduction in the level of *Mc4r* mRNA in the PVH. The new data are shown in Supplementary Fig. 4k.

3. The innovation and novelty of the paper lies in the attempt to manipulate PVHTrkB neurons based on their projection targets using several retrogradely transported Flp-recombinase viruses in combination with Flp-dependent Cre expression. Using this approach the authors present data demonstrating that *Ntrk2* in PVHTrkB → VMN and PVHTrkB → LPBN microcircuits is required for bodyweight control and normal feeding and the effect size is fairly equivalent (~10-12gm change over 8 weeks post-injection). In both cases, however, an important control group is not described: *Ntrk2* lox/lox mice injected with the flp-dependent Cre virus ALONE in the PVH. A control experiment is shown in Supplemental Figure 6 but this is in the striatum not the PVH; indeed some Cre activity appears to be present, as bGal staining (green cells) is detected in the image presented. Any independent Cre expression from this virus would recombine the *Ntrk2* lox/lox gene in the PVH and mimic the experiments conducted in Figure 3.

We co-injected AAV2-fDIO-mCherry-P2A-Cre with AAV2-GFP into the PVH and did not detect any leaky expression of mCherry. The new data are shown in Supplementary Fig. 8b. This result indicates that AAV2-fDIO-mCherry-P2A-Cre does not express Cre in the absence of FLPo. The green dots in Supplementary Fig. 8a are much smaller than cells positive for β-galactosidase in Fig. 6b. They could be stains for endogenous β-galactosidase or non-specific stains.

4. Given the critical importance of the non-overlapping microcircuitry between the PVH → VMN and PVH → PBN to a central theme of the manuscript, additional support for the independent nature of these circuits would be beneficial. For example, when retroAAV-Flp-2aGFP is injected into the PBN, presumably it is transported to the PVH cell body to generate Flp recombinase and GFP. Expression of GFP should be an adequate tracing tool for these neurons and will allow the investigators to determine if PVHTrkB neurons labeled at the terminals in the PBN also project to the VMN.

Injected AAV2retro-FLPo-T2A-GFP expresses GFP in the LPBN (injection site) as well as the PVH. Thus, it is not ideal to use GFP to examine the projection of PVH neurons into the LPBN. Instead, we examined mCherry expression to determine projections of these neurons. As shown in the new

Supplementary Fig. 9, we could detect axonal terminals of PVH^{→LPBN} neurons in the LPBN but not the VMH. This result further supports the conclusion that distinct PVH^{TrkB} neurons project to the LPBN and the VMH.

5. The paper would also be strengthened by combining a PVHTrkB->PBN and PVHTrkB->VMN manipulation to determine whether these two projection pathways effectively recapitulate the obesity phenotype seen with pan-PVH deletion of Ntrk2. Based on the data presented, each projection subset contributes 10-12gm of excess bodyweight which is quite close to the total body mass gained by panPVH deletion (~25gm; figure 3)

This is a very difficult experiment. The rate of successfully injecting viruses into six small brain structures in a mouse is very low. We did not carry out the experiment.

6. Given the novelty of the PVH to VMN satiety circuit, it is unclear why the investigators chose to demonstrate Fos changes associated with feeding only in the PVH->PBN circuit (Fig 7g,h).

We have performed the suggested experiment and added new data on Fos induction associated with feeding in the PVH^{TrkB->VMH} neurons in this revised manuscript (Fig. 8a-c). Refeeding also induced Fos expression in these neurons.

7. The fact that unilateral deletion of Ntrk2 also leads to obesity should be addressed in more detail. This is especially relevant given the fact that the loss of significant Ntrk2 from oxytocin neurons or from PVH->NTS projecting neurons has little effect on energy balance.

We have briefly discussed this result in both Result and Discussion sections in this revised manuscript.

Minor concerns:

In Figure 2, the description of NtrkfBZ/+ allele is confusing; I realize that it is referenced, but the authors would aid the reader in providing a description of the allele. The legend states that Figure 2A shows Ntrk2 deletion; however, in this figure bGal expression is a proxy for Ntrk2 expression. Also, representation of SimCre mediated Ntrk2 deletion in Fig 2B,C should show close ups of the hypothalamus in regions SimCre is expected to be active in order to allow the reader to more clearly visualize the deletion of Ntrk2.

We have added a description of the Ntrk2^{fBZ} allele. We have also revised Fig. 2b to include enlarged images of the PVH.

For most chemogenetic activation experiments, Fos expression is following administration of CNO is used to demonstrate that transduced neurons are activated by CNO. This may be a useful addition for this manuscript as a means to determine if PVH neurons neighboring those transduced by the 3Dq-DREADD are also being activated indirectly in response to CNO.

We found that 80% of Fos-expressing cells in the PVH were those that expressed hM3D(Gq)-mCherry after CNO administration (Supplementary Fig. 6). This new result indicates that the vast majority of neurons activated by CNO are hM3D(Gq)-expressing TrkB cells in the PVH.

In Figure 3d, a “hit score” is represented graphically, but never defined. What is a “hit score”? Also representative images of the PVH (Fig 3a2) should be provided in increased magnification, similar to that represented by the fluorescent image in Fig3a1.

We now define the “hit score” in the figure legend for Fig. 3d. We also have increased the magnification of in situ hybridization images.

The “proof of concept” experiment using CAV2-FlpO into the striatum and Flp-dependent mCherry-2Acre could be moved to supplemental data as its lengthy description detracts from the flow of the paper.

We think that the data are important for this manuscript and would prefer to keep the data in a main figure.

Reviewers' Comments:

Reviewer #1:

Remarks to the Author:

The authors had carefully addressed my comments. I only have a few very minor comments:

- 1) The methods of quantification used in Figure 1 and Suppl. Figure 1 need more detail. How many sections? How many animals? Any error bars? For transparency, the information should also be included in the Excel file provided with numbers.
- 2) The volumes and concentrations of CTB and amounts of green RetroBeads injections needs to be stated in the methods section.
- 3) Just want to confirm the coordinates of the injections. For example: the PVH injection DV was -5.4 mm? seems much deeper than the Paxinos atlas suggestion (-4.6-4.7). Was this in reference of the surface of the skull?

Reviewer #2:

Remarks to the Author:

Overall, the authors were very responsive to the previous review and addressed all of the issues raised. New RNAscope data showing the fidelity of the TrkB-td tomato reporter are included with the revised manuscript (Fig. 1a). They indicate that about 14% of TrkB- cells express the reporter. Could the authors comment as to how they think this ectopic expression of the reporter, albeit small, might impact interpretation of the co-expression studies of tdtomato and various PVN peptides?

Reviewer #3:

Remarks to the Author:

The authors have addressed all of my major concerns regarding the manuscript with additional experiments and specific controls. They have extended their characterization of overlapping PVH populations with Ntrk2 cells including Mc4R and Glp1R, cell types known to contribute to energy balance control. Assessment of PVH Ntrk2 to IPBN and VMN projecting cells in response to fasting/refeeding helps establish the contribution of these cells/pathways in basal physiology.

Reviewer #1 (Remarks to the Author):

The authors had carefully addressed my comments. I only have a few very minor comments:

1) The methods of quantification used in Figure 1 and Suppl. Figure 1 need more detail. How many sections? How many animals? Any error bars? For transparency, the information should also be included in the Excel file provided with numbers.

We have included section numbers and animal numbers in Methods. Because the animal number is one or two, we do not put error bars to the colocalization figures. Cell numbers are now included in the Source Data file.

2) The volumes and concentrations of CTB and amounts of green RetroBeads injections needs to be stated in the methods section.

We now describe the volumes in figure legends.

3) Just want to confirm the coordinates of the injections. For example: the PVH injection DV was -5.4 mm? seems much deeper than the Paxinos atlas suggestion (-4.6-4.7). Was this in reference of the surface of the skull?

We used the surface of the skull as the reference for DV. We now describe this in Methods.

Reviewer #2 (Remarks to the Author):

Overall, the authors were very responsive to the previous review and addressed all of the issues raised. New RNAscope data showing the fidelity of the TrkB-td tomato reporter are included with the revised manuscript (Fig. 1a). They indicate that about 14% of TrkB- cells express the reporter. Could the authors comment as to how they think this ectopic expression of the reporter, albeit small, might impact interpretation of the co-expression studies of tdtomato and various PVN peptides?

The reviewer might mean that about 14% of TrkB cells do not express the reporter. It is unlikely that mRNAs for TrkB and tdTomato fill the whole cytoplasm, especially in cells with low *Ntrk2* expression. Because a confocal image only captures fluorescence from a thin optical section of a cell, some cells in one confocal image could contain signals for one of two mRNAs even if the cells express both mRNAs.

Reviewer #3 (Remarks to the Author):

The authors have addressed all of my major concerns regarding the manuscript with additional experiments and specific controls. They have extended their characterization of overlapping PVH populations with *Ntrk2* cells including Mc4R and Glp1R, cell types known to contribute to energy balance control. Assessment of PVH *Ntrk2* to IPBN and VMN projecting cells in response to fasting/refeeding helps establish the contribution of these cells/pathways in basal physiology.

Thanks.